# LUNA: Efficient and Topology-Agnostic Foundation Model for EEG Signal Analysis

**Berkay Döner**[1]  **Thorir Mar Ingolfsson**[1]*  **Luca Benini**[1]  **Yawei Li**[1]
[1]Integrated Systems Laboratory, ETH Zürich, Switzerland

## Abstract

Electroencephalography (EEG) offers a non-invasive lens into human brain activity, but building large-scale models is hampered by *topological heterogeneity*: each public EEG data defines its own electrode layout, limiting generalization. We introduce **LUNA** (**L**atent **U**nified **N**etwork **A**rchitecture), a self-supervised foundation model that reconciles disparate electrode geometries while scaling linearly—not quadratically—with channel count. LUNA compresses multi-channel EEG into a fixed-size, topology-agnostic latent space via *learned queries* and cross-attention. Downstream transformer blocks then operate exclusively on this latent representation using patch-wise temporal self-attention, decoupling computation from electrode count. Pre-trained on TUEG and Siena ($> 21,000$ hours of raw EEG across diverse montages) using a masked-patch reconstruction objective, LUNA transfers effectively to four downstream tasks: abnormality detection, artifact rejection, slowing classification, and emotion recognition. It demonstrates highly competitive performance across several benchmarks, achieving state-of-the-art results on TUAR and TUSL, e.g., **0.921 AUROC** on TUAR, while reducing FLOPs by $300\times$ and trimming GPU memory use by up to $10\times$. Critically, these gains are consistent across all evaluated electrode configurations. Code is available at https://github.com/pulp-bio/biofoundation

## 1   Introduction

Electroencephalography (EEG) provides deep insight into brain activity without requiring invasive procedures, and plays a crucial role in clinical diagnostics, cognitive neuroscience, and human-computer interaction. In recent years, deep neural networks have significantly advanced EEG analysis, shifting from handcrafted pipelines to end-to-end learning systems [1]. Transformer-based models now rival traditional signal processing techniques by jointly modelling long-range temporal dynamics and cross-channel correlations [2, 3].

Despite this progress, *a fundamental bottleneck remains*: EEG corpora exhibit significant *topological heterogeneity*. Electrode count and placement vary widely across public and private datasets, making it difficult to transfer models across montages. This limitation manifests in pronounced performance degradation during cross-dataset evaluation. For example, motor-imagery decoders lose up to 14 percentage points (pp) in accuracy when transferring from PhysioNet to KU datasets [4], while state-of-the-art emotion-recognition models such as BIOT and MMM exhibit 13–15 pp drops between SEED and DEAP montages [5, 6]. Similarly, patient-to-patient transfer in stereotactic EEG (sEEG) remains an unsolved challenge, with naive models performing near chance without explicit spatial encoding [7].

Existing approaches offer limited solutions to this problem. Some train bespoke models for each montage, while others retain only shared electrodes—discarding up to **80%** of available data [8].

---

*Correspondence to {thoriri,yawli}@iis.ee.ethz.ch.

39th Conference on Neural Information Processing Systems (NeurIPS 2025).

More general approaches that flatten channels and time into long sequences incur quadratic self-attention complexity, $\mathcal{O}\big((S \cdot C)^2\big)$ where $S$ is the number of time segments and $C$ is the number of electrodes (channels), rapidly exhausting memory on dense caps [5]. These challenges underscore the need for a **single, montage-agnostic architecture that scales efficiently with electrode count**.

**LUNA** (**L**atent **U**nified **N**etwork **A**rchitecture) directly addresses this gap. Our key innovation is a topology-invariant encoder that maps arbitrary electrode layouts into a fixed latent space via learned queries and cross-attention. Temporal self-attention layers then operate exclusively on this latent space, decoupling computational cost from the number of electrodes. We pre-train LUNA using a masked-patch reconstruction objective on TUEG [9] and SIENA [10] (over *21,000* hours of raw EEG data), and fine-tune on four downstream benchmarks spanning abnormality and artifact detection, slowing classification, and emotion recognition.

The key contributions of this work are the following:

- **Topology-invariant encoder.** A learnt query / cross-attention module that projects arbitrary-sized channel sets into a fixed latent space.
- **Linear-in-channels complexity.** Patch-wise temporal attention that decouples FLOPs and memory from electrode count.
- **State-of-the-art accuracy-efficiency trade-off.** LUNA achieves strong results across a range of EEG benchmarks, demonstrating significant capabilities with balanced accuracies of **81.57%** on TUAB and **39.18%** on SEED-V [11], and AUROC scores of **0.921** on TUAR and **0.802** on TUSL, while reducing FLOPs by **300×** and GPU memory footprint by up to **10×** on high-density EEG recordings. Crucially, these gains hold across diverse electrode configurations, confirming LUNA's generalization capability.

## 2 Related Work

To contextualize our contributions, this section discusses relevant state-of-the-art methodologies that we will compare against. We focus on advancements in self-supervised learning for time series, the emergence of foundation models for physiological signals, and existing approaches to managing variable input structures, especially concerning topological heterogeneity in the EEG domain and computational efficiency.

### 2.1 Self-Supervised Learning Strategies in EEG

Foundation models for EEG primarily rely on self-supervised learning (SSL) to leverage large unlabeled datasets. Masked signal modeling is a dominant paradigm. BENDR [12] pioneered this for EEG by adapting masked prediction concepts from speech, applying a contrastive objective to predict masked convolutional features. Subsequent models refined this: BrainBERT [13] performs masked prediction on channel-independent spectrograms for intracranial electroencephalography (iEEG); EEGFormer [14] and LaBraM [15] predict vector-quantized (VQ) representations of masked patches, learning discrete codebooks; CBraMod [16] directly reconstructs masked raw signal patches. LUNA employs a similar masked reconstruction objective but applies it after projecting channel information into a unified latent space, requiring the decoder to reconstruct channel-specific details from this compressed representation.

### 2.2 Modeling Spatial Structure and Topology Variation in EEG

Capturing the spatial relationships between EEG channels is vital but complicated by varying electrode counts and layouts across datasets. Several strategies have been explored in the literature:
**Channel Independence:** Early approaches and models like BrainBERT [13] and EEGFormer [14] process each channel's data independently before potentially combining them later. While inherently handling varying channel numbers, this neglects early modeling of cross-channel interactions.
**Fixed-Topology Spatial Modeling:** Models like Brant [17] use dedicated spatial encoders alongside temporal ones but assume a consistent channel configuration, limiting cross-dataset generalization. Graph Neural Networks (GNNs) [18] explicitly model spatial relationships using a predefined adjacency graph, but require mechanisms to handle dynamically changing graph structures when topologies vary. LUNA avoids pre-defined graphs or fixed structures.

**Joint Spatio-Temporal Attention:** LaBraM [15] flattens channel and patch dimensions into one long sequence, allowing a standard Transformer to learn spatio-temporal dependencies simultaneously. However, this incurs $\mathcal{O}((S \cdot C)^2)$ complexity, scaling quadratically with both sequence length/patches (S) and channels (C). CBraMod [16] and CEReBrO [19] use alternating or parallel spatial and temporal attention mechanisms, reducing complexity to $\mathcal{O}(max(S^2, C^2))$ but still scaling quadratically with the dominant dimension. BIOT [5] uses linear attention after flattening, improving efficiency but potentially limiting modeling capacity. LUNA differs significantly by performing channel unification first before applying temporal attention with quadratic complexity only on the patch dimension and the much smaller latent dimension Q.

**Explicit Topology Mapping:** Some methods explicitly map varying topologies to a canonical representation. MMM [6] maps channels to predefined anatomical regions but relies on hand-engineered features (Differential Entropy) rather than raw signals. PopT [20] aggregates pre-computed channel-independent temporal features using 3D electrode coordinates. While achieving topology invariance, these methods are not fully end-to-end or rely on external information (regions). LUNA learns an end-to-end mapping from raw signals using learned queries without requiring pre-defined structures.

**Differentiable Channel Reordering.** Saeed et al. [21] also use learned attention to reorder/project heterogeneous channels into a fixed space, but under supervised training and without explicit 3D channel encodings. In contrast, LUNA is a self-supervised foundation model trained at scale, integrates explicit spatial (3D) information, and targets topology-agnostic *and* compute-efficient transfer across datasets and tasks.

### 2.3   Learned Queries and Efficient Attention for Set Abstraction

LUNA's core mechanism for topology unification draws inspiration from architectures designed for permutation-invariant processing of set-structured data. Set Transformer [22] introduced the concept of using a small set of learnable inducing points (queries) and an Induced Set Attention Block to summarize information from a larger input set via cross-attention, reducing the complexity from $\mathcal{O}(N^2)$ to $\mathcal{O}(M \cdot N)$. PerceiverIO [23] further developed this mechanism, demonstrating its power in creating a fixed-size latent bottleneck capable of handling diverse, variable-sized inputs across different modalities (images, text) and enabling flexible decoding via task-specific output queries.

LUNA adapts this principle specifically for EEG topology invariance. We treat the set of EEG channel features at a given time interval (patch) as the input set. By applying cross-attention between the channel features (as keys/values) and a small number (Q) of learned queries, LUNA projects the variable-channel input onto a fixed-size latent space ($\mathbb{R}^{Q \times E}$). This projection is permutation-invariant with respect to the input channels, thus achieving topology agnosticism. Furthermore, it improves computational efficiency, as the complexity of this step scales linearly with the number of channels. Where MMM relies on predefined regions (and hand-crafted features) and LaBraM/CBraMod rely on quadratic spatial attention after flattening space–time, LUNA first *unifies* variable channel sets with learned queries (with explicit 3D encodings) and *then* applies temporal attention on a fixed-size latent, yielding linear-in-$C$ unification and reduced temporal sequence length. This design specifically targets topology-agnostic scaling and inference efficiency within a self-supervised foundation-model framework, rather than combining prior components unchanged.

## 3   Methodology

Developing generalizable foundation models for EEG is hindered by two primary obstacles: the **topological heterogeneity** of EEG montages (varying channel counts and layouts) and the **computational complexity** of attention mechanisms. Standard models struggle with diverse input channel configurations, limiting data aggregation and generalizability. Furthermore, transformer-based approaches often face prohibitive $\mathcal{O}((C \cdot S)^2)$ or $\mathcal{O}(max(C^2, S^2))$, as discussed in the section 2.2, complexity when processing $C$ channels and $S$ temporal patches. This limits their applicability to high-density EEG or long recordings.

LUNA addresses these challenges using a smaller latent space. Firstly, Channel-Unification Module (Sec . 3.1) employs learned queries and cross-attention to project variable-channel features into a fixed-dimension latent space, achieving topology invariance. Secondly, by unifying channel information into a compact set of $Q$ queries ($Q \ll C$) before temporal processing, LUNA significantly reduces computational demands. This design enables efficient and scalable processing of heterogeneous EEG

data, paving the way for more robust foundation models. LUNA adopts an encoder-decoder architecture that transforms EEG signals from heterogeneous montages into a unified latent representation, enabling topology-agnostic modeling and efficient downstream decoding (Figure 1).

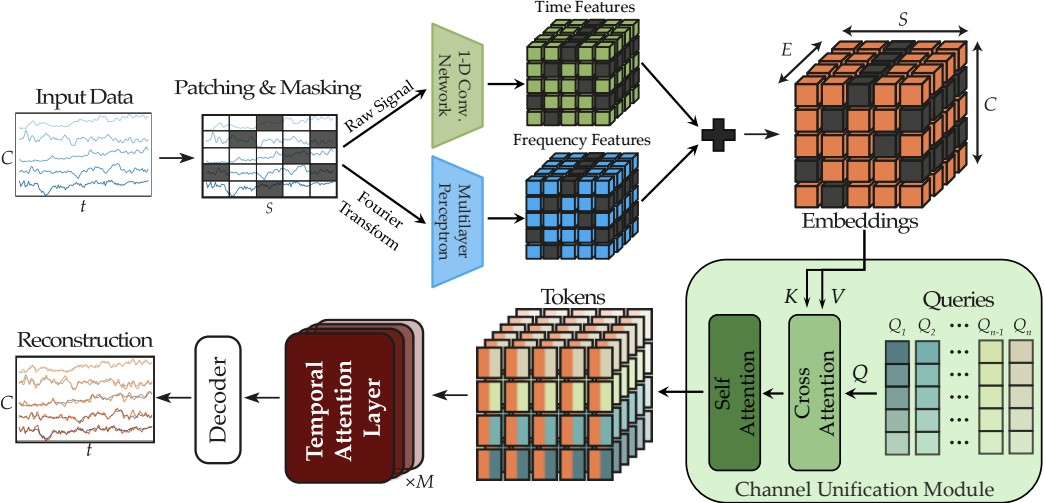

Figure 1: Overview of LUNA. EEG signals ($B \times C \times T$) are segmented into patches and embedded. Channel-Unification Module maps channel-wise features into a fixed-size latent space using learned queries ($Q$). Patch-wise Temporal Attention processes this latent sequence. The decoder generates task-specific outputs.

## 3.1 Encoder

The encoder comprises three key modules that transform the input EEG into a topology-agnostic latent representation: patch feature extraction, channel unification, and patch-wise temporal modeling.

**Patch Feature Extraction**  Given raw EEG $x \in \mathbb{R}^{B \times C \times T}$ (Batch $B$, Channels $C$, Time $T$), we segment each channel into $S = T/P$ non-overlapping temporal patches of size $P$. These patches are embedded via two parallel pathways:

**Temporal Embedding:** A 1D convolutional network (with GroupNorm [24], GELU [25]) encodes local temporal features similar to state-of-the-art methods such as LaBraM[15] and CBraMod [16], **Frequency Embedding:** The magnitude and phase from each patch's Fourier transform are projected through an MLP. These representations are summed to obtain patch features $x_{\text{features}}$.

**Channel Positional Encoding**  To encode electrode locations, we apply NeRF-inspired sinusoidal encoding [26] to normalized 3D electrode coordinates, followed by an MLP projection. This yields $\mathbf{E}_{\text{pos}} \in \mathbb{R}^{B \times C \times E}$, which is added to $x_{\text{features}}$.

During pre-training, a random subset of tokens is masked using a learnable embedding.

**Channel-Unification Module**  To handle varying channel counts ($C$) across recordings, we introduce a cross-attention module that maps patch-wise features into a fixed latent space. Specifically, Q learned queries $\mathbf{Q}_{\text{learn}} \in \mathbb{R}^{Q \times E}$, (learnable parameters *without* a batch dimension, initialized orthogonally) cross-attend to patch features. For attention, these queries are repeated across the $B \cdot S$ patch instances as $\tilde{\mathbf{Q}} \in \mathbb{R}^{(B \cdot S) \times Q \times E}$.

Let the input to this module be the tensor $\mathbf{X}_{\text{token}} \in \mathbb{R}^{B \times (C \cdot S) \times E}$, representing the spatially-aware features for $B$ samples, $S$ patches per channel, and feature dimension $E$. We first reshape this tensor to $\mathbf{X}' \in \mathbb{R}^{(B \cdot S) \times C \times E}$ to treat each patch instance across the batch independently while isolating the channel dimension for attention.

The cross-attention mechanism then computes the output representation $\mathbf{A}_{\text{out}} \in \mathbb{R}^{(B \cdot S) \times Q \times E}$:

$$\mathbf{A}_{\text{out}} = \text{MultiHeadAttention}(\tilde{\mathbf{Q}}, \mathbf{X}', \mathbf{X}') \qquad (1)$$

A feed-forward network (FFN) with residual connection refines the outputs, followed by $L$ Transformer encoder layers operating on the query dimension $Q$.

$$\mathbf{X}_{\text{unified}} = \text{TransformerEncoder}(\mathbf{A}_{\text{out}} + \text{FFN}(\mathbf{A}_{\text{out}})) \tag{2}$$

The result $\mathbf{X}_{\text{unified}} \in \mathbb{R}^{(B \cdot S) \times Q \times E}$ decouples further processing from the original electrode layout. *For clarity: learnable tensors (e.g., $\mathbf{Q}_{learn}$) omit the batch dimension; repetition occurs only at attention time (e.g., $\tilde{\mathbf{Q}}$).*

**Patch-wise Temporal Encoder** The unified representations are reshaped into temporal sequences $\mathbf{X}'_{\text{unified}} \in \mathbb{R}^{B \times S \times (Q \cdot E)}$. These are processed by a stack of Transformer encoder blocks with Rotary Positional Embeddings (RoPE) [27] to capture temporal dependencies efficiently. A key advantage of this encoding approach is that each of the $S$ temporal tokens in $\mathbf{X}'_{\text{unified}}$ now encapsulates richer, aggregated information from multiple input channels, rather than representing a single channel's segment. Furthermore, by not tokenizing each channel independently for temporal processing, the effective sequence length for the temporal Transformers is reduced from $S \cdot C$ to just $S$, leading to significant reductions in computational complexity and memory requirements.

$$E_{\text{out}} = \text{TemporalEncoder}(\mathbf{X}'_{\text{unified}})$$

## 3.2 Decoder

LUNA supports two decoding strategies depending on the task: reconstruction (pre-training) and classification (fine-tuning).

**Reconstruction Head (Pre-training)** For masked patch reconstruction, $C$ learned decoder queries $\mathbf{E}_{\text{dec}}^{\text{learn}} \in \mathbb{R}^{C \times E}$ (learnable parameters *without* a batch dimension) are repeated across the batch and patches as $\tilde{\mathbf{E}}_{\text{dec}} \in \mathbb{R}^{(B \cdot S) \times C \times E}$ and attend to reshaped $E_{\text{out}}$ via cross-attention, producing channel-specific representations $\mathbf{Z}_{\text{dec}} \in \mathbb{R}^{(B \cdot S) \times C \times E}$. $E_{\text{out}} \in \mathbb{R}^{B \times S \times (Q \cdot E)}$ is reshaped to be used as keys/values $\mathbf{K}, \mathbf{V} \in \mathbb{R}^{(B \cdot S) \times Q \times E}$ for attention. A linear projection $\phi : \mathbb{R}^E \to \mathbb{R}^P$ applied on $\mathbf{Z}_{\text{dec}}$ recovers the patch values. *Decoder queries are indexed by channel labels and can be reused across datasets when electrodes overlap; this reconstruction head is used only during pre-training.*

**Classification Head (Fine-tuning)** For downstream tasks, a single aggregation query $\mathbf{E}_{\text{agg}}^{\text{learn}} \in \mathbb{R}^{1 \times (Q \cdot E)}$ (no batch) is repeated across the batch as $\tilde{\mathbf{E}}_{\text{agg}} \in \mathbb{R}^{B \times 1 \times (Q \cdot E)}$ and attends to $E_{\text{out}}$ to produce a pooled representation, which is passed to an MLP for classification.

## 3.3 Training Objectives

LUNA is pre-trained with a masked reconstruction loss and an auxiliary query specialization loss.

**Reconstruction Loss** A Smooth L1 loss is applied to both masked and visible patches:

$$L_{rec} = \frac{1}{N_{\text{masked}}} \sum_{i \in M} \text{SmoothL1}(x_{\text{orig}_i}, x_{\text{recons}_i}) + \alpha \cdot \frac{1}{N_{\text{visible}}} \sum_{i \notin M} \text{SmoothL1}(x_{\text{orig}_i}, x_{\text{recons}_i})$$

and $\text{SmoothL1}(x, \hat{x}) = 0.5(x - \hat{x})^2$ if $|x - \hat{x}| < \beta$, else $\beta|x - \hat{x}| - 0.5\beta^2$, with $\beta = 1$.

**Affinity matrix.** Let $\mathbf{A}_{\text{attn}} \in \mathbb{R}^{B' \times H \times Q \times C}$ denote the cross-attention weights (queries $\to$ channels) from the channel-unification module, for $B'$ instances and $H$ heads. We define

$$\mathbf{A}_{\text{affinity}} = \frac{1}{H} \sum_{h=1}^{H} \mathbf{A}_{\text{attn}}[:, h, :, :] \in \mathbb{R}^{B' \times Q \times C},$$

i.e., attention weights averaged over heads, so that $(\mathbf{A}_{\text{affinity}})_{b', q, c}$ measures the affinity between query $q$ and channel $c$ for instance $b'$.

**Query Specialization Loss**   To promote diversity among queries, we penalize similarity in query-channel affinity matrices by minimizing the mean value of off-diagonal elements:

$$\mathcal{L}_{\text{spec}} = \frac{\lambda_{\text{spec}}}{B' \cdot Q \cdot (Q-1)} \sum_{b'=1}^{B'} \sum_{i=1}^{Q} \sum_{j=1, j \neq i}^{Q} \left( (\mathbf{A}_{\text{affinity}} \mathbf{A}_{\text{affinity}}^{T})_{b',i,j} \right)^2$$

## 4   Results

### 4.1   Experimental Setup

**Datasets**   We pre-train LUNA on a combined corpus of Temple University Hospital EEG Corpus (TUEG) [9] and the Siena Scalp EEG Database [10], spanning recordings with 20, 22, and 29 channels amounting to over 21,900 hours of EEG data (see Table 11). Downstream evaluations cover four diverse benchmarks: **TUAB** [9]: Abnormal EEG detection (binary classification), **TUAR** [9]: We follow prior work [28] and treat artifact detection as a *multiclass* (single-label) problem with five classes (one label per segment). Artifact detection (multi-class classification) **TUSL** [9]: Slowing event classification (4-class classification). **SEED-V** [11]: Emotion recognition (5-class classification), with unseen 62-channel topology. All subjects and recordings from the downstream evaluation datasets (TUAB, TUAR, TUSL, SEED-V) were strictly excluded from this pre-training set to ensure fair evaluation of generalization. For LUNA, the input EEG is segmented into patches, consisting of 40 timestamps. For most datasets, EEG recordings are sliced into non-overlapping 5-second segments to form individual training/evaluation samples. SEED-V dataset uses its default 1-second sample duration.

**Fine-tuning and Data Splits**   For the TUAB dataset, we use the official train-test split. As TUSL and TUAR lack official subject-wise test splits, we follow recent leading work (e.g., EEGFormer [14]) and adopt an 80%/10%/10% randomized sample-level split for train/val/test to allow direct, like-for-like comparison. We acknowledge that subject-independent splits are the gold standard for assessing clinical generalization and recommend them for future benchmark comparisons. For SEED-V, fifteen trials are divided equally into train, validation, and test sets for each session. For the TUAR dataset, we adopt a multiclass classification approach, restricting to 5 distinct artifact types in a single-label setting, similar to EEGFormer [14]. We optimize binary cross-entropy loss for TUAB and cross-entropy loss for other datasets. We report the mean and standard deviation of results obtained across three different random seeds.

**Preprocessing**   We apply a minimal, standardized preprocessing pipeline to all EEG data. Signals are first bandpass filtered between 0.1 Hz and 75 Hz. A notch filter (50Hz or 60Hz) is applied to remove power-line interference. All signals are then resampled to 256 Hz. For TUEG, TUAB, TUAR, and TUSL datasets we construct a bipolar ("double-banana") montage by differencing predefined longitudinal electrode pairs provided in the dataset documentation; the full list of channel pairs used is given in Appendix A.7. Siena and SEED-V are processed in unipolar format. Finally, each channel within each sample is normalized using z-score normalization.

**Computational Environment**   All experiments were conducted on a cluster of eight NVIDIA A100 GPUs, using Python 3.11.6 and PyTorch 2.4.1 with CUDA 12.1. Training utilizes 'bf16' mixed-precision. Detailed hyperparameters for pre-training and fine-tuning are provided in Appendix A.3.

**Baselines and Variants**   We compare against state-of-the-art supervised and self-supervised methods, including transformer-based architectures such as LaBraM [15], CBraMod [16], EEGFormer [14], and BIOT [5]. LUNA is evaluated in three configurations: Base (7M), Large (43M), and Huge (311M parameters). Model size is increased by expanding the depth of the Patch-wise Temporal Encoder, the hidden embedding dimension $E$, and the number/size of queries $Q$ in the Channel-Unification Module. Key architectural settings are detailed in Appendix A.1.

### 4.2   Downstream Task Performance

**Abnormal EEG Detection (TUAB)**   LUNA delivers competitive performance on TUAB (Table 1). LUNA-Huge achieves AUROC of 0.8957 and AUPR of 0.9029, surpassing most self-supervised

baselines and approaching large-scale models like LaBraM and CBraMod. Notably, LUNA maintains strong performance while offering substantial efficiency and topology-agnostic benefits relative to strong self-supervised and large-scale baselines.

Table 1: Performance comparison on TUAB abnormal EEG detection.

| Model | Size | Bal. Acc. (%) ↑ | AUC-PR ↑ | AUROC ↑ |
|---|---|---|---|---|
| *Supervised Models* | | | | |
| SPaRCNet [29] | 0.8M | 78.96 ± 0.18 | 0.8414 ± 0.0018 | 0.8676 ± 0.0012 |
| ContraWR [30] | 1.6M | 77.46 ± 0.41 | 0.8421 ± 0.0140 | 0.8456 ± 0.0074 |
| CNN-Transformer [31] | 3.2M | 77.77 ± 0.22 | 0.8433 ± 0.0039 | 0.8461 ± 0.0013 |
| FFCL [32] | 2.4M | 78.48 ± 0.38 | 0.8448 ± 0.0065 | 0.8569 ± 0.0051 |
| ST-Transformer [33] | 3.2M | 79.66 ± 0.23 | 0.8521 ± 0.0026 | 0.8707 ± 0.0019 |
| *Self-supervised Models* | | | | |
| BENDR [12] | 0.39M | 76.96 ± 3.98 | - | 0.8397 ± 0.0344 |
| BrainBERT [13] | 43.2M | - | 0.8460 ± 0.0030 | 0.8530 ± 0.0020 |
| EEGFormer-Base [14] | 2.3M | - | 0.8670 ± 0.0020 | 0.8670 ± 0.0030 |
| BIOT [5] | 3.2M | 79.59 ± 0.57 | 0.8692 ± 0.0023 | 0.8815 ± 0.0043 |
| EEG2Rep [34] | - | 80.52 ± 2.22 | - | 0.8843 ± 0.0309 |
| FEMBA-Huge [35] | 386M | 81.82 ± 0.16 | 0.9005 ± 0.0017 | 0.8921 ± 0.0042 |
| CEReBrO [19] | 85.15M | 81.67 ± 0.23 | 0.9049 ± 0.0026 | 0.8916 ± 0.0038 |
| LaBraM-Base [15] | 5.9M | 81.40 ± 0.19 | 0.8965 ± 0.0016 | 0.9022 ± 0.0009 |
| LaBraM-Huge [15] | 369.8M | **82.58 ± 0.11** | 0.9204 ± 0.0011 | **0.9162 ± 0.0016** |
| CBraMod [16] | 69.3M | 82.49 ± 0.25 | **0.9221 ± 0.0015** | 0.9156 ± 0.0017 |
| **LUNA-Base** | 7M | 80.63 ± 0.08 | 0.8953 ± 0.0016 | 0.8868 ± 0.0015 |
| **LUNA-Large** | 43M | 80.96 ± 0.10 | 0.8986 ± 0.0005 | 0.8924 ± 0.0010 |
| **LUNA-Huge** | 311.4M | 81.57 ± 0.11 | 0.9029 ± 0.0014 | 0.8957 ± 0.0011 |

**Artifact and Slowing Detection (TUAR and TUSL)** LUNA delivers state-of-the-art results on TUAR and TUSL (Table 2). LUNA-Huge achieves AUROC 0.921 on TUAR, outperforming FEMBA-Large and other methods. On TUSL, LUNA-Huge reaches AUROC 0.802, the highest among all compared models.

Table 2: Performance comparison on TUAR (artifact detection) and TUSL (slowing event classification).

| Model | Size | TUAR | | TUSL | |
|---|---|---|---|---|---|
| | | AUROC ↑ | AUC-PR ↑ | AUROC ↑ | AUC-PR ↑ |
| *Supervised Models* | | | | | |
| EEGNet [36] | - | 0.752 ± 0.006 | 0.433 ± 0.025 | 0.635 ± 0.015 | 0.351 ± 0.006 |
| EEG-GNN [18] | - | 0.837 ± 0.022 | 0.488 ± 0.015 | 0.721 ± 0.009 | 0.381 ± 0.004 |
| GraphS4mer [37] | - | 0.833 ± 0.006 | 0.461 ± 0.024 | 0.632 ± 0.017 | 0.359 ± 0.001 |
| *Self-supervised Models* | | | | | |
| BrainBERT [13] | 43.2M | 0.753 ± 0.012 | 0.350 ± 0.014 | 0.588 ± 0.013 | 0.352 ± 0.003 |
| EEGFormer-Base [14] | 2.3M | 0.847 ± 0.014 | 0.483 ± 0.026 | 0.713 ± 0.010 | **0.393 ± 0.003** |
| EEGFormer-Large [14] | 3.2M | 0.852 ± 0.004 | 0.483 ± 0.014 | 0.679 ± 0.013 | 0.389 ± 0.003 |
| FEMBA-Base [35] | 47.7M | 0.900 ± 0.010 | **0.559 ± 0.002** | 0.731 ± 0.012 | 0.289 ± 0.009 |
| FEMBA-Large [35] | 77.8M | 0.915 ± 0.003 | 0.521 ± 0.001 | 0.714 ± 0.007 | 0.282 ± 0.010 |
| **LUNA-Base** | 7M | 0.902 ± 0.011 | 0.495 ± 0.010 | 0.767 ± 0.023 | 0.301 ± 0.003 |
| **LUNA-Large** | 43M | 0.918 ± 0.003 | 0.505 ± 0.010 | 0.771 ± 0.006 | 0.293 ±0.021 |
| **LUNA-Huge** | 311.4M | **0.921 ± 0.011** | 0.528 ± 0.012 | **0.802 ± 0.005** | 0.289 ±0.008 |

**Emotion Recognition on Unseen Montage (SEED-V)** The SEED-V benchmark tests generalization to a novel 62-channel montage, distinct from pre-training data. Results in Table 3 show that while LUNA effectively operates on this unseen topology, its performance (e.g., Bal. Acc.) lags behind leading methods like CBraMod by 2-3 pp. This suggests a trade-off inherent in LUNA's design: while its query-based unification enables efficient, topology-agnostic processing across common montage

variations (as demonstrated on TUAB/TUAR/TUSL), generalizing zero-shot to vastly different, high-density layouts remains challenging, possibly due to positional encoding constraints. Despite this gap, LUNA shows positive scaling from Base to Large models, underscoring its potential.

Table 3: Performance comparison on SEED-V emotion recognition (5-classes).

| Model | Size | Bal. Acc. (%) ↑ | Cohen's Kappa ↑ | Weighted F1 ↑ |
|---|---|---|---|---|
| *Supervised Models* | | | | |
| SPaRCNet [29] | 0.79M | $0.2949 \pm 0.0078$ | $0.1121 \pm 0.0139$ | $0.2979 \pm 0.0083$ |
| ContraWR [30] | 1.6M | $0.3546 \pm 0.0105$ | $0.1905 \pm 0.0188$ | $0.3544 \pm 0.0121$ |
| CNN-Transformer [31] | 3.2M | $0.3678 \pm 0.0078$ | $0.2072 \pm 0.0183$ | $0.3642 \pm 0.0088$ |
| FFCL [32] | 2.4M | $0.3641 \pm 0.0092$ | $0.2078 \pm 0.0201$ | $0.3645 \pm 0.0132$ |
| ST-Transformer [33] | 3.5M | $0.3052 \pm 0.0072$ | $0.1083 \pm 0.0121$ | $0.2833 \pm 0.0105$ |
| *Self-supervised Models* | | | | |
| BIOT [5] | 3.2M | $0.3837 \pm 0.0187$ | $0.2261 \pm 0.0262$ | $0.3856 \pm 0.0203$ |
| LaBraM-Base [15] | 5.8M | $0.3976 \pm 0.0138$ | $0.2386 \pm 0.0209$ | $0.3974 \pm 0.0111$ |
| CBraMod [16] | 14M | $\mathbf{0.4091 \pm 0.0097}$ | $\mathbf{0.2569 \pm 0.0151}$ | $\mathbf{0.4101 \pm 0.0108}$ |
| **LUNA-Base** | 7M | $0.3730 \pm 0.0098$ | $0.1831 \pm 0.0103$ | $0.3389 \pm 0.0091$ |
| **LUNA-Large** | 43M | $0.3918 \pm 0.0066$ | $0.2073 \pm 0.0045$ | $0.3586 \pm 0.0013$ |
| **LUNA-Huge** | 311.4M | $0.3900 \pm 0.0096$ | $0.2037 \pm 0.0103$ | $0.3506 \pm 0.0047$ |

Unless noted, we report mean $\pm$ s.d. over matched seeds and focus on effect sizes and confidence intervals. Formal significance tests are summarized in Appendix A.9

### 4.3 Computational Efficiency

**LUNA achieves substantially better calling efficiency** compared to full and alternating attention models. As shown in Figure 2a, LUNA's patch-wise attention enables thousands of temporal patches without the quadratic cost faced by LaBraM. Likewise, Figure 2b shows that LUNA maintains near-constant compute cost when channel count increases, outperforming CBraMod's $\mathcal{O}(C^2)$ scaling for dense EEG recordings. These results confirm that LUNA decouples inference cost from input montage, making it well-suited for long recordings or high-density EEG scenarios. We also consider BIOT (linear attention) as an efficiency-oriented baseline; detailed FLOPs/memory scaling versus LUNA is provided in Appendix A.1.3. Although many public datasets use $\sim$20–30 channels, research/clinical systems often employ 64–256 channels and longer windows; LUNA's $O(C)$ unification and reduced temporal sequence length enable such regimes where quadratic spatial attention becomes impractical.

### 4.4 Ablation Studies

*Choice of Q.* We explore the trade-off between the number of queries $Q$ and embedding size $E$ under a fixed $Q \cdot E$ budget; see Appendix A.6. We validate the impact of LUNA's key design choices on TUAB and TUAR (Table 4).

**Learned Queries vs. Fixed Regions** Replacing learned queries with predefined spatial regions (similar to MMM [6]) yields small AUROC changes ($-0.004$ to $-0.006$), within seed variation. We therefore emphasize the *practical* advantages of learned queries—data-driven flexibility without anatomy-specific priors—rather than a statistically significant metric gain (Appendix A.9).

**Query Specialization Loss** Removing the specialization loss results in modest AUROC changes (-0.003 to -0.006), again on the order of the reported variation, with small mixed effects on AUC-PR. We retain this loss for its *regularizing role*: it encourages a diverse, non-redundant set of spatial filters (see Fig. 4), which is desirable for robustness in challenging artifact conditions.

**Frequency Features** Ablating frequency embeddings leads to the largest drop (up to -0.012 AUROC), indicating a more consistent contribution complementary to temporal features.

### 4.5 Latent Space Analysis

**Pre-trained Representations** t-SNE visualizations (Figure 3) reveal that even before fine-tuning, LUNA's encoder captures task-relevant structure. Normal and abnormal EEGs form separate clusters in TUAB, while artifact classes are partially separated in TUAR, demonstrating effective pre-training.

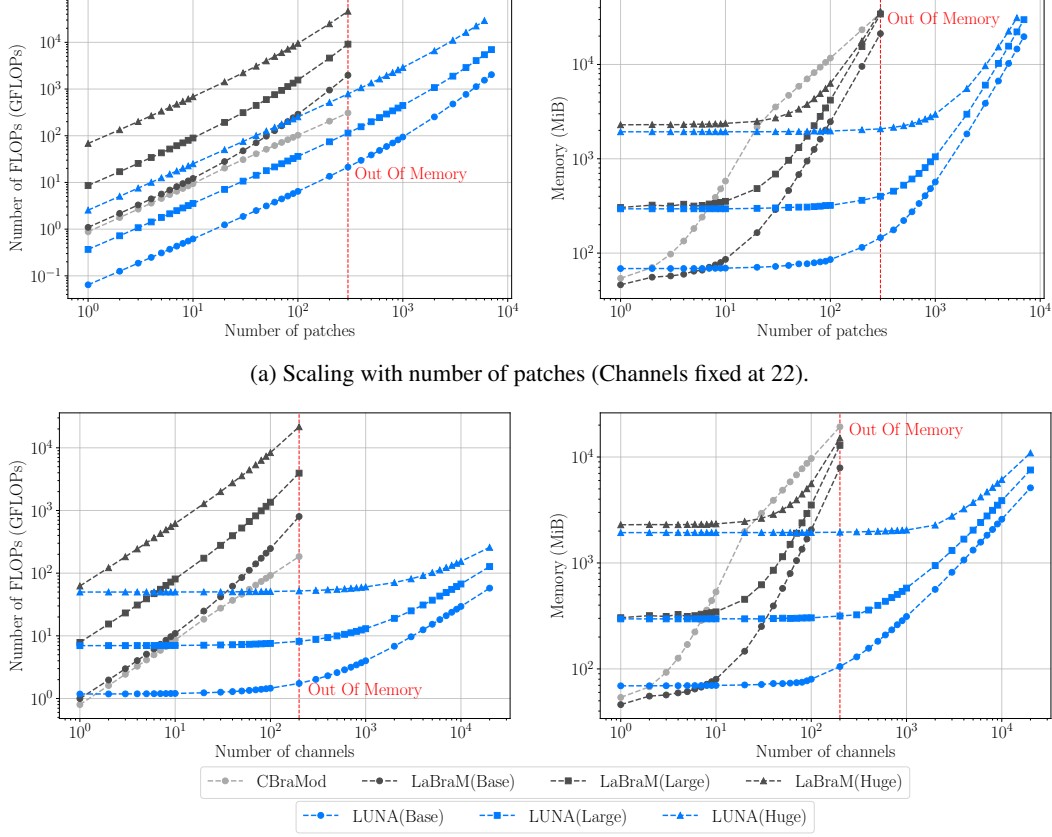

(a) Scaling with number of patches (Channels fixed at 22).

(b) Scaling with number of channels (Patches fixed at 20).

Figure 2: Computational cost scaling of LUNA and baseline models. (a) FLOPs and Memory usage vs. number of patches. (b) FLOPs and Memory usage vs. number of channels. LUNA demonstrates significantly better efficiency and scalability, especially compared to full attention (LaBraM), and favorable scaling compared to alternating attention (CBraMod) due to the fixed latent query space. CBraMod has a variable sized decoder based on the number of patches and channels; therefore, its model size as well as its resource usage grows rapidly. *FLOPs are measured with `fvcore`'s `FlopCountAnalysis` over 50 random inputs, including encoder+decoder, with window length $T$, patch size $P$, and reported as GFLOPs per forward pass.*

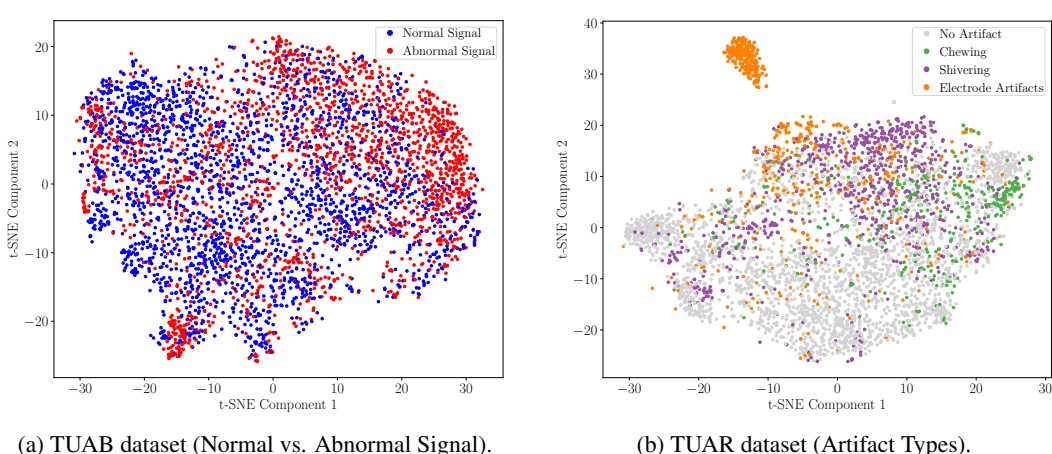

(a) TUAB dataset (Normal vs. Abnormal Signal).

(b) TUAR dataset (Artifact Types).

Figure 3: t-SNE of LUNA-Base embeddings on downstream datasets before fine-tuning.

### 4.6 Learned Query Specialization Visualization

**Query Specialization** Visual analysis of the learned queries (Figure 4) highlights their role in topology-agnostic representation. Queries exhibit distinct spatial profiles: some are localized (e.g., frontal regions), while others aggregate broader signals. This emergent specialization confirms that cross-attention learns flexible, data-driven basis functions for spatial unification.

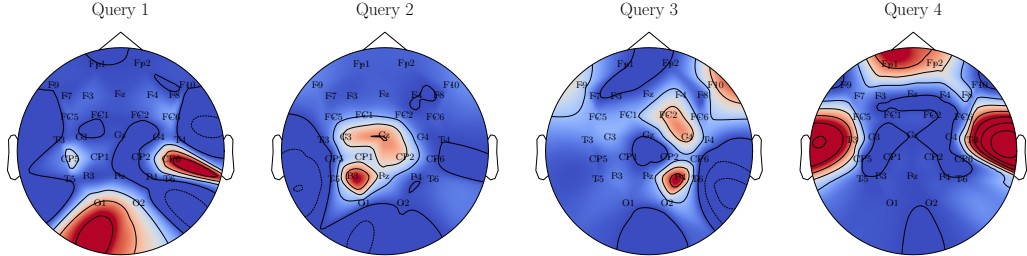

Figure 4: Visualization of the attention patterns of queries in LUNA-Base on Siena [10] topology.

Table 4: Ablation study results (LUNA-Base) on TUAB and TUAR datasets.

| Model Configuration | TUAB AUROC | TUAB AUC-PR | TUAR AUROC | TUAR AUC-PR |
|---|---|---|---|---|
| LUNA-Base (Full Model) | $0.887 \pm 0.002$ | $0.895 \pm 0.002$ | $0.902 \pm 0.011$ | $0.495 \pm 0.010$ |
| *Unification Module:* | | | | |
| - Region-based Attention | $0.883 \pm 0.001$ (↓ 0.004) | $0.892 \pm 0.002$ (↓ 0.003) | $0.896 \pm 0.001$ (↓ 0.006) | $0.509 \pm 0.006$ (↑ 0.014) |
| *Other Components:* | | | | |
| - w/o Query Specialization Loss | $0.884 \pm 0.003$ (↓ 0.003) | $0.892 \pm 0.002$ (↓ 0.003) | $0.895 \pm 0.005$ (↓ 0.007) | $0.498 \pm 0.010$ (↑ 0.003) |
| - w/o Frequency Features | $0.876 \pm 0.012$ (↓ 0.011) | $0.883 \pm 0.005$ (↓ 0.012) | $0.893 \pm 0.011$ (↓ 0.009) | $0.490 \pm 0.011$ (↓ 0.005) |

## 5 Conclusion

We introduced **LUNA**, a self-supervised foundation model designed to address the challenge of topological heterogeneity in EEG analysis. By leveraging learned queries and cross-attention, LUNA unifies recordings with diverse electrode layouts into a fixed latent space, enabling montage-agnostic modeling. Through extensive experiments across abnormality detection, artifact recognition, slowing classification, and emotion recognition, we demonstrate that LUNA matches or surpasses state-of-the-art performance while offering substantial efficiency gains in FLOPs and memory usage. Critically, these benefits hold across all evaluated electrode configurations.

While LUNA achieves strong results, especially on heterogeneous montages, our analysis also reveals limitations. Performance on SEED-V suggests sensitivity to unseen channel topologies, likely stemming from reliance on positional encodings learned during pre-training. Addressing this limitation, through enhanced spatial generalization strategies or hybrid learned/geometric embeddings, is an important direction for future work.

More broadly, this work highlights the promise of topology-agnostic latent representations for scalable EEG modeling. Future extensions include exploring unified models across EEG and invasive modalities (e.g., sEEG, ECoG), integrating domain-specific priors (e.g., neurophysiological constraints), and adapting LUNA for real-time inference scenarios. Beyond technical advancements, the development of efficient, topology-invariant EEG models like LUNA could enhance neurological diagnostics and research accessibility. However, careful attention must be paid to mitigating risks such as algorithmic bias and ensuring patient data privacy for deployment. Future work should integrate ethical concerns alongside technical improvements. Pre-training montage diversity is limited (three dominant layouts); future work will incorporate multi-dataset pre-training and randomized channel dropout to improve generalization to unseen, dense and sparse montages.

## Acknowledgments and Disclosure of Funding

This project is supported by the Swiss National Science Foundation under the grant number 193813 (PEDESITE project). This work was supported by the ETH Future Computing Laboratory (EFCL), financed by a donation from Huawei Technologies. This work was supported by a grant from the Swiss National Supercomputing Centre (CSCS) under project ID lp12 on Alps.

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

# A Appendix

This appendix provides supplementary details that complement the main paper: additional efficiency analyses, reporting notes for statistical testing, small implementation clarifications, and extra qualitative comparisons.

## A.1 Model Architecture Details

The following tables show the hyperparameter setup for the pre-training and the downstream fine-tuning for LUNA.

### A.1.1 Hyperparameters for pre-training

Table 5: Hyperparameters for EEG pre-training.

| Hyperparameters | | LUNA-Base | LUNA-Large | LUNA-Huge |
|---|---|---|---|---|
| Temporal Encoder | Input channels | {1,8,8} | {1,16,16} | {1,32,32} |
| | Output channels | {16,16,16} | {24,24,24} | {32,32,32} |
| | Kernel size | | {20,3,3} | |
| | Stride | | {10,1,1} | |
| | Padding | | {9,1,1} | |
| Patch size | | | 40 | |
| Transformer encoder layers | | 8 | 10 | 24 |
| Number of queries | | 4 | 6 | 8 |
| Query size | | 64 | 96 | 128 |
| Hidden size | | 256 | 576 | 1024 |
| MLP size | | 1024 | 2304 | 4096 |
| Attention head number | | 8 | 12 | 16 |
| Batch size per GPU | | 2040 | 2040 | 720 |
| Total batch size | | 8160 | 8160 | 11520 |
| Peak learning rate | | | 1.25e-4 | |
| Minimal learning rate | | | 2.5e-7 | |
| Learning rate scheduler | | | Cosine | |
| Optimizer | | | AdamW | |
| Adam $\beta$ | | | (0.9,0.98) | |
| Weight decay | | | 0.05 | |
| Total epochs | | | 60 | |
| Warmup epochs | | | 10 | |
| Loss type | | | Smooth-L1 | |
| Non-masked region loss coefficient | | | 0.05 | |
| Query specialization loss coefficient | | | 0.8 | |
| Gradient clipping | | | 1 | |
| Mask ratio | | | 0.5 | |
| Precision | | | bf16-mixed | |

### A.1.2 Hyperparameters for downstream fine-tuning

Table 6: Hyperparameters for downstream fine-tuning.

| Hyperparameters | Values |
|---|---|
| Batch size per GPU | 512 |
| Peak learning rate | 1e-4 |
| Minimal learning rate | 5e-6 |
| Learning rate scheduler | Cosine |
| Optimizer | AdamW |
| Adam $\beta$ | (0.9,0.999) |
| Weight decay | 0.05 |
| Total epochs | 50 |
| Early stopping patience | 10 |
| Warmup epochs | 5 |
| Drop path | 0.1 (B/L) 0.2 (H) |
| Layer-wise learning rate decay | 0.5 (B) 0.8 (L/H) |
| Label smoothing (multi-class classification) | 0.1 |

### A.1.3 Complexity Analysis

The computational complexity of key attention stages and a comparison with alternatives are shown in 7 and 8.

Table 7: Complexity Breakdown of LUNA Encoder Stages.

| Stage | Input Shape | Complexity |
|---|---|---|
| Channel-Unification Module (Cross-Attn) | $(B \cdot S) \times C \times E$ | $O(B \cdot S \cdot Q \cdot C \cdot E)$ |
| Query Self-Attention | $(B \cdot S) \times Q \times E$ | $O(B \cdot S \cdot Q^2 \cdot E)$ |
| Patch-wise Attention Encoder (Self-Attn) | $B \times S \times (Q \cdot E)$ | $O(B \cdot S^2 \cdot Q \cdot E)$ |

Table 8: Attention Complexity Comparison.

| Method | Bottleneck Complexity |
|---|---|
| LUNA (Latent Space Attention) | $O(B \cdot S^2 \cdot Q \cdot E)$ *or* $O(B \cdot S \cdot Q \cdot C \cdot E)$ |
| Full-Attention (e.g., LaBraM) | $O(B \cdot S^2 \cdot C^2 \cdot E)$ |
| Alternating Attention (Patches, e.g., CBraMod) | $O(B \cdot S^2 \cdot C \cdot E)$ |
| Alternating Attention (Channels, e.g., CBraMod) | $O(B \cdot S \cdot C^2 \cdot E)$ |

**BIOT vs. LUNA scaling.** Tables 9–10 report GFLOPs and peak activation memory across varying patch and channel counts.

### A.2 Dataset and Preprocessing Details

**Datasets Used** We use publicly available EEG datasets, provided in 11.

### A.3 Experimental Settings

**Pre-training** LUNA is pre-trained using a masked patch reconstruction task. Key hyperparameters are listed in 5.

**Computational Resources** Experiments were conducted using NVIDIA A100 GPUs. Pre-training took approximately 1 day on 8 GPUs for the base and large models and 16 GPUs for the huge model.

Table 9: Scaling with patch count (GFLOPs and MiB per forward).

| Model | #Patches | FLOPs (G) | Memory (MiB) |
|---|---|---|---|
| BIOT | 2000 | 1143 | 3534 |
| LUNA-Base | 2000 | 253 | 1835 |
| LUNA-Large | 2000 | 1073 | 2969 |
| LUNA-Huge | 2000 | 6570 | 5549 |
| BIOT | 3000 | 1714 | 5231 |
| LUNA-Base | 3000 | 478 | 3873 |
| LUNA-Large | 3000 | 1886 | 6041 |
| LUNA-Huge | 3000 | 11035 | 9685 |
| BIOT | 4000 | 2286 | 6931 |
| LUNA-Base | 4000 | 768 | 6678 |
| LUNA-Large | 4000 | 2884 | 10265 |
| LUNA-Huge | 4000 | 16286 | 15344 |

Table 10: Scaling with channel count (GFLOPs and MiB per forward).

| Model | #Channels | FLOPs (G) | Memory (MiB) |
|---|---|---|---|
| BIOT | 6000 | 3117 | 9270 |
| LUNA-Base | 6000 | 18 | 1571 |
| LUNA-Large | 6000 | 43 | 2410 |
| LUNA-Huge | 6000 | 112 | 4198 |
| BIOT | 7000 | 3637 | 10791 |
| LUNA-Base | 7000 | 20 | 1826 |
| LUNA-Large | 7000 | 49 | 2777 |
| LUNA-Huge | 7000 | 122 | 4680 |
| BIOT | 8000 | 4156 | 12307 |
| LUNA-Base | 8000 | 23 | 2069 |
| LUNA-Large | 8000 | 55 | 3139 |
| LUNA-Huge | 8000 | 133 | 5163 |

Table 11: Summary of Datasets Used.

| Dataset | # Subjects | # Samples (Train/Val/Test or Total) | Hours of Recordings | # Channels | Montage Used |
|---|---|---|---|---|---|
| TUEG (Pre-train) | 14,987 | 15,686,874 (Total) | 21,787.32 | 20 or 22 | Bipolar |
| Siena (Pre-train) | 14 | 101,520 (Total) | 141.0 | 29 | Unipolar |
| TUAB | 2,329 | 591,357 / 154,938 / 74,010 | 1,139.31 | 22 | Bipolar |
| TUAR | 213 | 49,241 / 5,870 / 5,179 | 83.74 | 22 | Bipolar |
| TUSL | 38 | 16,088 / 1,203 / 2,540 | 27.54 | 22 | Bipolar |
| SEED-V | 15 | 43,328 / 43,360 / 31,056 | 32.70 | 62 | Unipolar |

## A.4 Additional Quantitative Results

**Training Curves** The pre-training loss curves for LUNA-Base are shown in 5. The reconstruction loss drops shows and initial plateau then drops slowly over the epochs, while the query specialization shows a jump and then a slow decrease, indicating more orthogonal query usage over time. The initial drop of the query specialization might be due to a trivial case where a query attends to only one channel. The queries learn to attend to their own specialized areas afterwards while covering all the channels in the input.

## A.5 Additional Visualizations

**Reconstruction Examples** Figures 6, 7, 8 show examples of the model reconstructing masked patches (gray regions) for inputs with 20, 22, and 29 channels, respectively. The reconstructions capture the underlying signal trend and demonstrate robustness across different topologies seen during pre-training.

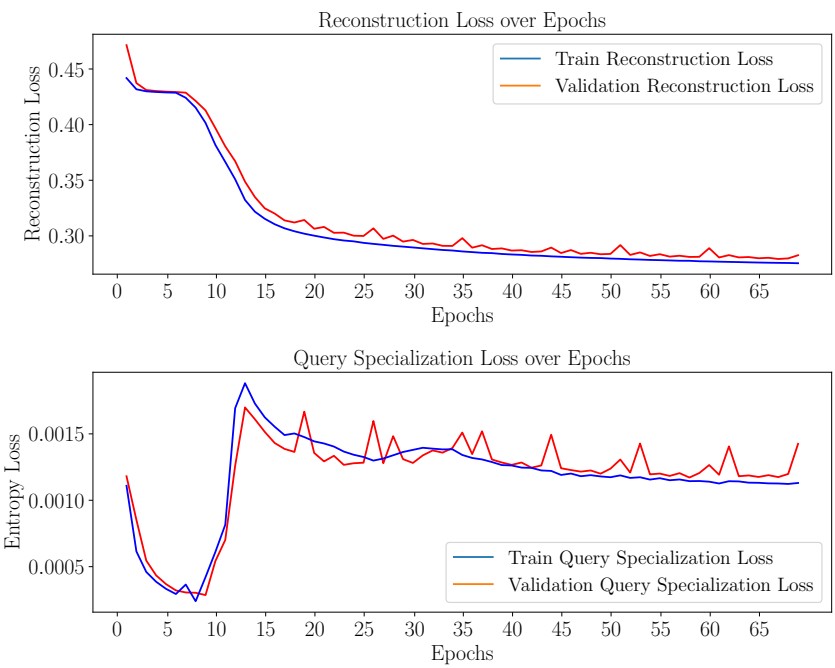

Figure 5: Loss curves during pre-training for LUNA-Base (Reconstruction and Query Specialization Loss).

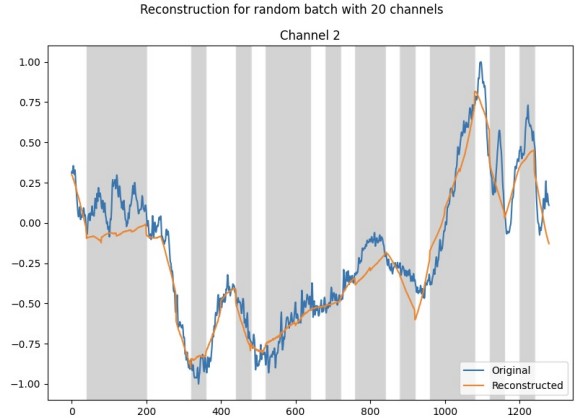

Figure 6: Example reconstruction on input with 20 channels (masked regions in gray).

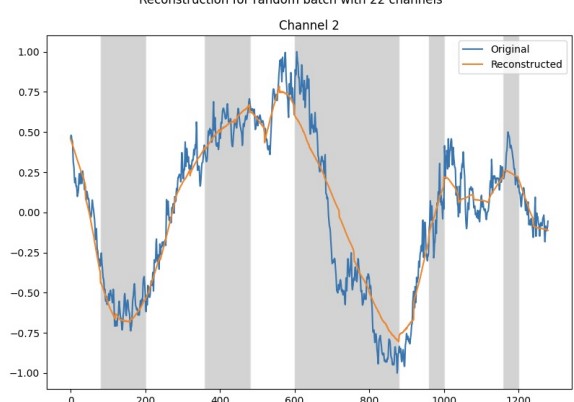

Figure 7: Example reconstruction on input with 22 channels (masked regions in gray).

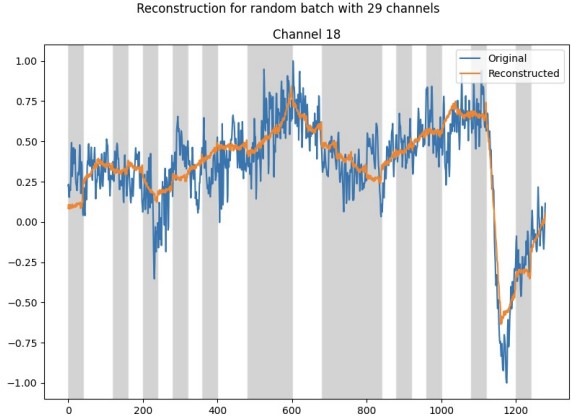

Figure 8: Example reconstruction on input with 29 channels (masked regions in gray).

## A.6 Trade-off between $Q$ and $E$

*Observation.* Increasing $Q$ at the expense of $E$ degrades performance; too few queries can also bottleneck capacity. A balanced $Q \times E$ (e.g., $4 \times 64$) works well under the same latent budget.

## A.7 Bipolar montage electrode pairs

We use the following longitudinal pairs to construct the bipolar montage for TUEG, TUAR, TUSL, and TUAB (left/right symmetric sets):

- Fp1–F7, F7–T3, T3–T5, T5–O1,   Fp2–F8, F8–T4, T4–T6, T6–O2 T3–C3, C3–CZ
- Fp1–F3, F3–C3, C3–P3, P3–O1,   Fp2–F4, F4–C4, C4–P4, P4–O2 CZ–C4, C4–T4

## A.8 Edge deployment reference

Typical low-power edge SoCs used in wearables/IoT offer single-digit to few dozen MB of RAM and on-chip/SiP compute in the 10–100 GOPS range. Under these constraints, LUNA-Base ($\sim$7M params; $\sim$14 MB at 16-bit) and its measured GFLOPs per window (Fig 2b) fit comfortably within real-time budgets, whereas quadratic-in-$C$ spatial attention and larger activation footprints in some baselines make deployment more challenging at higher channel counts.

## A.9 Significance Testing

Table 12: LUNA-Base variants with fixed $Q \cdot E$=256.

| Variant ($Q \times E$) | $Q$ | $E$ | TUAB AUROC | TUAB AUPRC | TUAR AUROC | TUAR AUPRC | TUSL AUROC |
|---|---|---|---|---|---|---|---|
| $4 \times 64$ | 4 | 64 | 0.887 | 0.895 | 0.902 | 0.495 | 0.767 |
| $2 \times 128$ | 2 | 128 | 0.885 | 0.890 | 0.885 | 0.501 | 0.759 |
| $8 \times 32$ | 8 | 32 | 0.884 | 0.892 | 0.899 | 0.505 | 0.766 |
| $16 \times 16$ | 16 | 16 | 0.874 | 0.881 | 0.866 | 0.487 | 0.757 |

Unless noted, we report mean $\pm$ s.d. over matched seeds. For ablations we ran two-sided paired $t$-tests across seeds for specific comparisons requested by reviewers. For the TUAR AUROC comparison of the model with vs. without the query specialization loss, the paired $t$-test yielded $p$=0.2136, i.e., not statistically significant at $\alpha$=0.05. Observed AUROC deltas across ablations were small (absolute $\leq 0.01$).

Table 13: Paired $t$-test on TUAR AUROC for the specialization-loss ablation.

| Comparison | Mean $\Delta$ (w/o $-$ full) | $p$ |
|---|---|---|
| w/o specialization vs. full | $-0.007$ | 0.2136 |

**Practical tolerance.** We adopt a pragmatic tolerance of $\pm 0.01$ AUROC for considering two variants practically equivalent on these datasets; all reported ablations fall within this band.

### A.10 Effect of Specialization Loss on Query Maps

Figure 9 depicts spatial attention maps of the $Q$ queries when the specialization loss is *removed*. In this setting, two queries converge to coarse lateralized patterns (left and right longitudinal chains), while the remaining queries display broad, overlapping support over fronto–central and midline sites with weaker focal peaks, indicating partial redundancy and gaps in complementary coverage. Overall, the maps exhibit higher overlap and reduced distinctiveness across queries compared to the model trained with the loss, where query maps are more complementary and less overlapping Fig. 4).

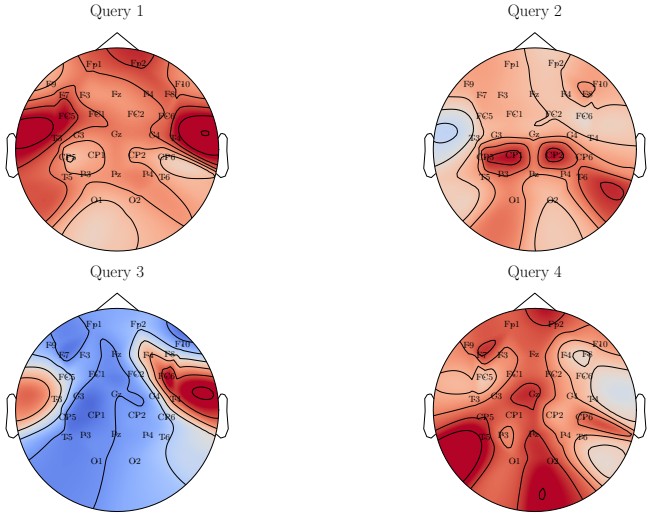

Figure 9: Query attention visualization without specialization loss.

### A.11 t-SNE of Raw Frequency Features vs. LUNA Features

We compute per-segment frequency features (magnitude/phase statistics per band, averaged across channels) and compare 2D t-SNE embeddings to those of LUNA's latent features. Raw features exhibit less separation beyond clear artifacts on TUAR; LUNA features show tighter clustering aligned with labels.

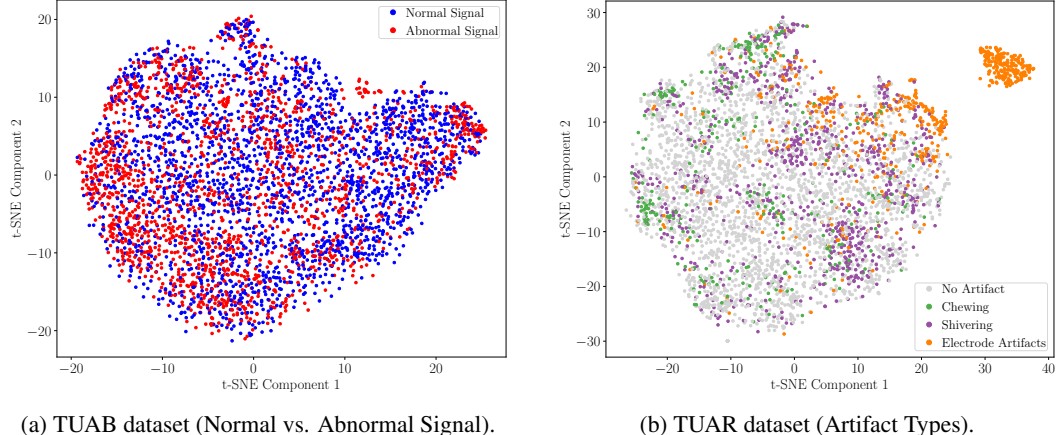

(a) TUAB dataset (Normal vs. Abnormal Signal).

(b) TUAR dataset (Artifact Types).

Figure 10: t-SNE of raw features on downstream datasets before fine-tuning.

