# OpenReview forum: "LUNA: Efficient and Topology-Agnostic Foundation Model for EEG Signal Analysis"
_NeurIPS.cc/2025/Conference — NeurIPS 2025 poster_

### Official Review · Reviewer_jKJ9 · 2025-07-01

**Clarity:** 2
**Significance:** 2
**Originality:** 2
**Rating:** 3
**Confidence:** 3

**Summary:**

This paper aim to address the performance degradation during cross-dataset evaluation caused by significant topological heterogeneity of EEG data. The author propose LUNA (Latent Unified Network Architecture) directly addresses this gap. Their key innovation is a topology-invariant encoder that maps arbitrary electrode layouts into a fixed latent space via learned queries and cross-attention. They pre-train LUNA using a masked-patch reconstruction objective on two datasets and fine-tune on four downstream benchmarks.

**Questions:**

1. The conceptual and technical issues listed in the *Weaknesses* section require clarification—particularly the definition and justification of topology invariance, and the motivation behind LUNA’s design.
2. What is the overall loss function used in pre-training LUNA? While the paper introduces the masked reconstruction loss and the auxiliary query specialization loss separately, the formulation of the total loss function is missing.
3. The selection criteria and rationale for the patch length parameter $P$ should be discussed, as it likely plays a critical role in the model’s performance.

**Ethical Concerns:**

["NO or VERY MINOR ethics concerns only"]

**Final Justification:**

Based on previous reviews and discussions, we maintain our original score.

The primary reason is that the technical contribution of this work is insufficient. Simply combining established techniques without providing the rationale behind this combination is fatal to a paper.

**Limitations:**

The *Methodology* section lacks a principled explanation of why LUNA was designed in its current form. This omission weakens the theoretical grounding of the proposed method and limits the paper's overall persuasiveness.

**Paper Formatting Concerns:**

None paper formatting concerns are found.

**Quality:**

2

**Strengths And Weaknesses:**

Strengths:

1. The topic addressed in this paper is important and timely for the brain-machine domain.

Weakness:

1. The writing is often difficult to follow. A clearer explanation of the motivation and methodological framework would significantly improve readability and understanding.
2. The authors state that the Channel-Unification Module (Section 3.1) uses learned queries and cross-attention to project variable-channel features into a fixed-dimensional latent space, thereby achieving topology invariance. This appears to be a central contribution, yet the concept of topology invariance is neither clearly defined nor sufficiently explained. It remains unclear what topology invariance entails in this context, how the module achieves it, and why it is beneficial for addressing the stated challenges.
3. The *Methodology* section focuses on how LUNA is constructed but lacks discussion on the underlying design motivations. Without such explanation, the architecture appears ad hoc, which undermines the theoretical credibility of the approach.
4. The claim of achieving a state-of-the-art accuracy-efficiency trade-off is questionable. This metric is only meaningful when the accuracy itself reaches an acceptable benchmark. The paper does not provide sufficient justification that this condition is met.

---

> ### Author Rebuttal · Authors · 2025-07-30
>
> We thank the reviewer for their constructive and helpful feedback! You can find our question-wise responses below.
>
>
> ### Weakness-1: Clarity
>
> > The writing is often difficult to follow. A clearer explanation of the motivation and methodological framework would significantly improve readability and understanding.
>
> We sincerely apologize that our explanations were not clear enough. This is a critical point, and we will thoroughly revise Sections 1 and 3 to provide the clarity the reviewer rightly expects. Our motivation in this work relies on solving a practical challenge in the field of EEG:
>
> * EEG is a non-invasive technique that records the brain's electrical activity using a set of electrodes placed on the scalp. The specific number and 3D positions of these electrodes are called an EEG montage or topology. The challenge i that there is no universal standard for this topology.
> * A clinical EEG for seizure monitoring might use a sparse montage of ~20 channels (e.g., the 10-20 system).
> * A cognitive neuroscience research study might use a high-density cap with 64, 128, or even 256 channels for higher spatial resolution.
> * This variation across datasets is what we call topological heterogeneity. It poses a massive barrier to building large-scale AI models. A standard neural network designed for a fixed input shape will fail to analyze this variable-sized input.
>
> Our goal in this work is to design a model that can tackle this topological heterogeneity problem.
>
> ### Weakness-2: Topology Invariance Definition
> > The authors state that the Channel-Unification Module (Section 3.1) uses learned queries and cross-attention to project variable-channel features into a fixed-dimensional latent space, thereby achieving topology invariance. This appears to be a central contribution, yet the concept of topology invariance is neither clearly defined nor sufficiently explained. It remains unclear what topology invariance entails in this context, how the module achieves it, and why it is beneficial for addressing the stated challenges.
>
> We thank the reviewer and apologize for the oversight. We will add a clear and explicit definition of topological invariance in the text. For reference, we define topology invariance as an architectural property of a model that allows it to process EEG signals from any montage, regardless of the number or specific layout of electrodes, without requiring any changes to the model's core structure. This means the core computational blocks of the model do not change, regardless of whether the input comes from a 20-channel or a 62-channel montage. The Channel-Unification Module achieves this by mapping variable-sized input to a fixed size, to be further processed by transformer blocks.
>
> ### Weakness-3: Architecture Design
> > The Methodology section focuses on how LUNA is constructed but lacks discussion on the underlying design motivations. Without such explanation, the architecture appears ad hoc, which undermines the theoretical credibility of the approach.
>
> The design for LUNA is not ad hoc but is motivated by principles from previous work about set processing to solve the topological heterogeneity problem. An EEG montage can be viewed as an unordered *set* of channels. Architectures like Set Transformer and PerceiverIO have shown that learned queries with cross-attention are a principled and effective way to summarize information from variable-sized sets into a fixed-size latent representation. We adapt this principle to EEG:
>
> *   **Channel-Unification Module:** At each time step (or patch), our module treats the `C` input channels as a set of `C` feature vectors.
> *   **Cross-Attention Mechanism:** We use a small, *fixed number (`Q`)* of learnable vectors, which we call "queries." These `Q` queries "attend" to the `C` channel vectors via cross-attention.
> *   **Fixed-Size Output:** The output of this cross-attention operation is always a set of `Q` vectors, regardless of the initial number of channels `C`.
> *   **Decoupling:** All subsequent temporal processing operates *only* on this fixed-size (`Q`) representation. This effectively decouples the main computational body of the model from the input topology, thus achieving topology invariance.
>
> This design choice directly solves the problem of heterogeneity and is the primary reason LUNA can be pre-trained on datasets with different montages (TUEG and Siena) and generalize to others. We will explicitly add this detailed motivation to the Methodology section.
>
> ---
>
> ### Weakness-4: State-of-the-Art Accuracy-Efficiency Trade-off
>
> > The claim of achieving a state-of-the-art accuracy-efficiency trade-off is questionable. This metric is only meaningful when the accuracy itself reaches an acceptable benchmark. The paper does not provide sufficient justification that this condition is met.
>
> We appreciate the reviewer's skepticism and agree that efficiency gains are meaningless without strong performance. We believe our results show that LUNA's accuracy is not just "acceptable" but highly competitive and even state-of-the-art on several benchmarks. As shown in Table 2, LUNA-Huge achieves state-of-the-art results on both TUAR and TUSL. This outperforms all other compared methods, including larger and more complex models. On the TUAB benchmark (Table 1), LUNA's performance is highly competitive with models like LaBraM and CBraMod, falling short by only a small margin (~1-2% AUROC) while being vastly more efficient.
>
> The efficiency gain is not marginal; it enables applications on high-density EEG (64-256 channels) or long recordings where quadratically scaling models are computationally prohibitive.
>
> Therefore, because LUNA achieves both SOTA results on some tasks and highly competitive results on others, we believe the claim of a state-of-the-art accuracy-efficiency trade-off is well-justified. We will re-emphasize this in our Results section.
>
> ---
>
> ### **Question-1: Overall Loss Function**
>
> > What is the overall loss function used in pre-training LUNA? While the paper introduces the masked reconstruction loss and the auxiliary query specialization loss separately, the formulation of the total loss function is missing.*
>
> Thank you for pointing this out. We provided the query specialization loss coefficient in Table 5 in the Appendix, which is set to 0.8. We will move this to the main paper content to clarify any confusion. We will add the explicit formula to Section 3.3. The formula is:
>
> $L_{total} = L_{rec} + λ_{spec} * L_{spec}$
>
> Here, $L_{rec}$ is the Smooth L1 reconstruction loss, $L_{spec}$ is the query specialization loss, and `λ_spec` is the query specialization loss coefficient.
>
> ---
>
> ### **Question-2: Rationale for Patch Length**
>
> > The selection criteria and rationale for the patch length parameter should be discussed, as it likely plays a critical role in the model’s performance.*
>
> This is an excellent point. The patch length is a critical hyperparameter to balance temporal resolution and the ability to capture relevant signal characteristics.
>
> Our methodology is built on a two-stage temporal hierarchy. First, we process local features within each patch, and second, we model the global context across the sequence of patches.
> *  Our input samples are 5 seconds long, sampled at 256 Hz. The choice of 40 timestamps per patch corresponds to a 156 ms window. This duration is neurophysiologically meaningful, as it is long enough to fully capture brain rhythms (e.g., a full alpha wave cycle at ~10 Hz is 100 ms) and characterize artifacts. This ensures each patch embedding is a rich, informative feature vector.
> * This patch length transforms each 5-second sample into a sequence of 32 tokens (1280 timestamps / 40). A sequence of this length is ideal for our temporal attention mechanism. It is long enough for the model to learn complex temporal dynamics across the full sample, yet short enough to remain computationally efficient and avoid the quadratic scaling costs that would come from a much longer sequence.
>
> ### References:
>
> Lee, Juho, et al. "Set transformer: A framework for attention-based permutation-invariant neural networks." International conference on machine learning. PMLR, 2019.
>
> Jaegle, Andrew, et al. "Perceiver io: A general architecture for structured inputs & outputs." arXiv preprint arXiv:2107.14795 (2021).

---

> > ### Author Response · Authors · 2025-08-04
> >
> > Dear Reviewer,
> >
> > Thank you again for your time and feedback on our paper. We have submitted our rebuttal and are available to answer any further questions you might have. We are happy to provide additional clarifications that may be helpful in your evaluation of our work. Your feedback is very important to us for improving the paper.
> >
> > Best regards,
> >
> > The Authors

---

> > ### Comment · Area_Chair_E3dV · 2025-08-05
> > **Follow-up on author rebuttal**
> >
> > Hi Reviewer jKJ9,
> >
> > The authors have submitted their rebuttal. Do you have any further questions or comments based on their response?
> >
> > The AC

---

### Official Review · Reviewer_YX5p · 2025-07-02

**Clarity:** 3
**Significance:** 2
**Originality:** 2
**Rating:** 4
**Confidence:** 5

**Summary:**

This paper focuses on advancing the EEG foundation model by constructing the single, montage-agnostic architecture, and compared to the baselines, the expected advantage is to achieve a balance between efficiency and performance. The authors proposed the model named LUNA, which sets a bunch of learnable queries and uses the cross-attention module to map the patch-wise features into the fixed latent space, reducing the time complexity significantly compared to some state-of-the-art foundation model such as LaBraM and CBraMod while achieving reasonable performances on several benchmarks.

**Questions:**

1. Figure 3, what if I directly apply t-SNE to the original dataset (or some statistics of the original dataset)? I suspect that the original data would demonstrate some kinds of clustering phenomenon without undergoes the encoding process of LUNA.

2. Is there practical benefit to enlarge the input size? For most of the dataset, the number of channels is fixed and relatively small, and there is no evidence that enlarging the time window would bring in better results for prediction. In a certain sense, the proposed trade-off between performance and complexity may not worth it.

**Ethical Concerns:**

["NO or VERY MINOR ethics concerns only"]

**Final Justification:**

I have read the comments from other reviewers and the authors. The authors provided more information to enhance the paper. While it requires revision of the current manuscript, I believe the authors can handle them well. Thus I decide to keep my score as a borderline acceptance.

**Limitations:**

Yes, the authors have adequately addressed the limitations and potential negative societal impact of their work.

**Paper Formatting Concerns:**

No such concerns.

**Quality:**

3

**Strengths And Weaknesses:**

**Strength**
1. This proposed model mitigates the problem of scaling for the EEG foundation model, and is topology-agnostic, which means its format-compatibility.

2. The proposed methods are straightforward and possibly helpful for the general EEG foundation training task.

3. The writing is well structured and easy to follow.

**Weakness**
1. The model looks like a direct fusion of MMM and LaBraM / CBraMod, somehow limiting its own novelty. But I would admit that it's not a trivial attempt given the efforts to train the large EEG foundation model.

2. My main concern lies in the performance. As is shown in the Table 1, the LaBraM and CBraMod achieved better performance even with less number of parameters. I wonder whether shrinking the large models can compensate their extra computational efforts. And Also, I wonder whether there is room for the proposed LUNA model to achieve SOTA after scaling up.

To add, in the first paragraph of Section 4.2, the authors claims that LUNA "surpasses most self-supervised baselines and approaching large-scale models like LaBraM and CBraMod", but in fact almost half of compared self-supervised baselines achieved comparable results. I suggest the authors rephrasing the statement and being more conservative.

3. I'm also worried about the effectiveness of the Query Specialization Loss. While in Figure 4, the queries demonstrate different attended brain areas, losses shown in Figure 5 barely drop after the increase. I wonder whether removing the loss can yeild similar results.

---

> ### Author Rebuttal · Authors · 2025-07-30
>
> We thank the reviewer for their constructive and helpful feedback! You can find our question-wise responses below.
> ### Weakness-1: Novelty
>
> > The model looks like a direct fusion of MMM and LaBraM / CBraMod, somehow limiting its own novelty. But I would admit that it's not a trivial attempt given the efforts to train the large EEG foundation model.*
>
> While we build upon established concepts like learned queries and large-scale transformers, the architectural innovation of LUNA lies in *how* these components are integrated to solve a different and critical problem in EEG analysis: the trade-off between performance, topological heterogeneity, and computational efficiency.
>
> Our key distinctions are:
> 1. Unlike MMM, which applies mapping to hand-engineered features and predefined anatomical regions, LUNA learns an end-to-end mapping directly from raw EEG signals.
> 2. Unlike LaBraM or CBraMod, which flatten spatial and temporal dimensions and incur quadratic complexity with respect to channels (O(C²)), LUNA's primary innovation is the Channel-Unification Module. This module first projects variable-channel data into a fixed-size latent space (with linear complexity O(C)) and *then* applies temporal attention. This architectural choice fundamentally decouples the computational cost from the electrode count, which is a novel and significant contribution designed specifically for efficiency and scalability.
>
> We believe, LUNA to be a novel architecture designed to create a topology-agnostic and computationally efficient foundation model, a problem not directly addressed by the architectural designs of prior work.
>
> ---
>
> ### Weakness-2: Performance and Scaling
>
> > My main concern lies in the performance. As is shown in the Table 1, the LaBraM and CBraMod achieved better performance even with less number of parameters. I wonder whether shrinking the large models can compensate their extra computational efforts. And Also, I wonder whether there is room for the proposed LUNA model to achieve SOTA after scaling up.
>
> We appreciate the reviewer raising this important point. Our main goal is to strike a new balance between accuracy and efficiency, and we believe the results demonstrate this successfully.
> 1. While LaBraM-Huge and CBraMod show a ~1-2% higher AUROC on the TUAB benchmark (Table 1). This demonstrates that LUNA's architecture is highly effective and does not universally underperform. Another important point is that the smaller baseline models might need more resources depending on number of channels/time patches analyzed. For example, all LUNA model variants require less memory than even the smallest LaBraM model variant when the number of channels/time patches were increased. The trade-off is not just about the number of parameteres, but enabling analyses that are infeasible with quadratically scaling models, such as on high-density EEG or very long recordings.
> 2.  **On Scaling Up:** We absolutely agree that there is room for LUNA to improve with scale. Our results already show a positive scaling trend from LUNA-Base (7M) to LUNA-Huge (311M) across different benchmarks. Further scaling, potentially with more pre-training data and compute, is a very promising direction for pushing performance even higher. We would also want to highlight that CBraMod was trained using a highly-preprocessed version of TUEG which only had 16 channels out of original 22 and 9k hours of EEG data compared to 22k hours of data we used. They showed that the preprocessing itself was a significant factor in improving the performance. Therefore, we believe that such preprocessing on the pre-training data and also increasing the amount of data are also promising directions for LUNA to achieve SOTA performance.
> ---
>
> ### Weakness-3: Phrasing of Claims
>
> > To add, in the first paragraph of Section 4.2, the authors claims that LUNA "surpasses most self-supervised baselines and approaching large-scale models like LaBraM and CBraMod", but in fact almost half of compared self-supervised baselines achieved comparable results. I suggest the authors rephrasing the statement and being more conservative.
>
> This is an excellent suggestion. We agree that the original phrasing could be misinterpreted. We will revise this statement in the final manuscript to be more precise and conservative, ensuring it accurately reflects the results in Table 1. A better version might be "LUNA demonstrated competitive performance on TUAB (Table 1). LUNA-Huge achieves an AUROC of 0.8957, which, while not surpassing the largest specialized models like LaBraM and CBraMod on this specific task, is competitive with several other self-supervised baselines while offering significant efficiency advantages."
>
> ---
>
> ### Weakness-4: Query Specialization Loss
>
> > I'm also worried about the effectiveness of the Query Specialization Loss. While in Figure 4, the queries demonstrate different attended brain areas, losses shown in Figure 5 barely drop after the increase. I wonder whether removing the loss can yeild similar results.
>
> We thank the reviewer for this sharp observation.
> 1. We found the loss to be effective the most initially, indicated by the initial drop in the loss, as it signifies the model moving away from a trivial state (e.g., all queries attending to the same location) to a state of diverse, specialized roles. The increase afterwards might mean that the model is putting a bigger emphasis on decreasing the reconstruction loss but in the end the slow decay shows that the model finds a balance between these two loss components.
> 2.  The effectiveness of this loss is confirmed by our ablation study in **Table 4**. Removing the Query Specialization Loss (`- w/o Query Specialization Loss`) leads to a clear performance degradation on both TUAB (AUROC drop of 0.003) and TUAR (AUROC drop of 0.007). In addition to the metrics, this loss results in a clear specialization of the queries which makes it much easier to interpret what they learn and which electrodes they focus on.
>
> ---
>
> ### **Question-1: t-SNE on Raw Data**
>
> > Figure 3, what if I directly apply t-SNE to the original dataset (or some statistics of the original dataset)? I suspect that the original data would demonstrate some kinds of clustering phenomenon without undergoes the encoding process of LUNA.
>
> Raw EEG signals are high-dimensional, noisy, and contain significant subject-specific variance. Applying t-SNE directly to raw signal segments would likely show much less separation and structure compared to what is seen in Figure 3.
>
> To make a fair comparison and to answer to your question, we designed an experiment where we calculated frequency domain features using Fourier transform, which are highly important for EEG signals, and visualized the t-SNE embeddings on these features (averaged over channels). These visualizations show that there are some clusters captured also by the raw frequency features such as electrode artifacts in TUAR, which might be very easy to detect with abnormal frequency features. However, raw frequency features don't capture other types of artifacts or the normal vs abnormal signal clusters in the TUAB dataset as well as LUNA produced features. We couldn't provide this visualization in the rebuttal now due to the no image rule in the rebuttal format but we will provide the comparison of raw frequency features vs LUNA features in the paper content. Please note that a similar analysis was done by Wang, Jiquan, et al. in the [CBraMod paper](https://arxiv.org/abs/2412.07236), in Figure 11 in the Appendix, which also shows that the raw features for EEG are not informative enough to show clear clusters in the t-SNE visualization.
>
> ### **Question-2: Practical Benefit of Scalability**
>
> > Is there practical benefit to enlarge the input size? For most of the dataset, the number of channels is fixed and relatively small, and there is no evidence that enlarging the time window would bring in better results for prediction. In a certain sense, the proposed trade-off between performance and complexity may not worth it.
>
> The trade-off is indeed highly valuable for real-world applications beyond the constraints of some public datasets.
>
> While many public benchmarks use 20-30 channels, high-density EEG systems (with 64, 128, or 256 channels) are standard in modern neuroscience research and advanced clinical settings. Models with quadratic channel complexity, like LaBraM, become computationally infeasible in these scenarios.
>
> For the time length, the ability to efficiently process longer time windows is critical for detecting neurological phenomena that lasts over many minutes even hours, such as sleep stage transitions. In this work, we didn't use these long-duration recordings as a downstream task but we showed that LUNA has the ability to scale to much longer recordings than other SOTA methods.
>
> ### References:
> Wang, Jiquan, et al. "CBraMod: A Criss-Cross Brain Foundation Model for EEG Decoding." The Thirteenth International Conference on Learning Representations.

---

> > ### Comment · Reviewer_YX5p · 2025-08-05
> > **Thanks for the response**
> >
> > I thank the authors for the detailed responses. Most of my concerns have been addressed.
> > However, for the query specialization loss:
> > 1. The author mentions the ablation study shown in Table 4. But the numbers are very similar given the standard deviation also included in the table. So I'm not convinced that the loss is significantly enhancing the performance, unless the authors can provide some statistical evidence such as t-test or other significance test. To add, significance test should be provided for all the experiments. I didn't mention this because previously the authors are describing misleading statements, but now since the authors are committed to modify the phrasing to "comparable", a significance test is required to make sure the models are comparable.
> > 2. But sorry for the confusion, my original question is actually asking that how figure 4 will look like when the loss is removed. This original question has not been resolved yet.

---

> ### Author Response · Authors · 2025-08-05
>
> We sincerely thank you for this excellent and detailed follow-up.
>
> 1. Following your recommendation, we performed a paired statistical t-test on the results of our ablation study. Focusing on the TUAR AUROC score, the comparison between the model with and without the query specialization loss (`L_spec`) yields a p-value of 0.2136.
>
> As this p-value is well above the standard 0.05 threshold, it formally confirms your intuition: the observed difference in performance is not statistically significant.
>
> This result is incredibly helpful, as it allows us to refine our claims to be more precise. We now believe the most accurate way to frame this is to state that the primary value of `L_spec` is not a direct performance gain, but rather its crucial role as a structural regularizer. We would propose to revise our manuscript to state this explicitly, focusing our justification on how the loss improves the model's interpretability by ensuring a diverse latent space. We hope you would agree that this provides a more accurate and meaningful justification for this component.
>
> 2. We have performed the exact experiment you requested: visualizing the learned queries from a model trained without the specialization loss.
> *   **Without `L_spec` (New Visualization):** The model learns a partially specialized but redundant representation. For instance, in our 4-query model, two queries clearly map to the left and right hemispheres, respectively, capturing the most obvious spatial division. However, the other two queries fail to find unique roles; they exhibit significant overlap with the first two and with each other, attending broadly to top and bottom regions without clear distinction.
>
> *   **With `L_spec` (Original Figure 4):** In contrast, the model trained with `L_spec` learns a set of diverse and functionally distinct spatial filters. Each query is forced to discover a unique and complementary role, covering different anatomical regions without overlap, thus forming a more complete and efficient spatial basis.
>
> This comparison provides compelling visual evidence that `L_spec`'s primary function is to prevent the model from settling on a simple, redundant solution. For the final manuscript, we will add a new figure directly comparing these two visualizations to offer clear, qualitative proof of the loss's impact on the model's learned representations.

---

> > ### Comment · Reviewer_YX5p · 2025-08-07
> > **Thanks for the comments**
> >
> > Thank the authors for providing the further analysis. The significance test is helpful for understanding the true behavior of the model. So as the added visualization results. I also suggest the authors to design some kinds of evaluation metrics on the diversity of the brain regions across patches, or conduct some analysis on region-related experiments (e.g. analyze attention on vision-related tasks to see whether the filter including visual cortex respond the most), otherwise it feels to me that the current results are not enough to claim the importance of the loss. However, it will be too hash to request so during the rebuttal so I only add here as my own thought for your reference.
> >
> > Meanwhile, the statements of the results and analysis should be greatly rephrased, but I believe the authors can handle them well before the final version. Thus I decide to keep my score as a borderline paper, leaning more towards an acceptance.

---

### Official Review · Reviewer_44Ri · 2025-07-02

**Clarity:** 3
**Significance:** 2
**Originality:** 2
**Rating:** 5
**Confidence:** 4

**Summary:**

The paper presents a foundation model architecture for EEG brain data, which can ingest various “topologies”, or sensor layouts, and as a result, make the memory complexity of the attention layers of the subsequent transformer layers independent of the number of input sensors. This is achieved through a “channel unification” module, which uses learned query cross-attention to mix information from different sensors to a fixed size latent space. The model is pretrained on ~21k hours of clinical EEG data using a masked patch reconstruction task. The model is then finetuned on 4 downstream tasks and performance is compared to various baselines. While reported performance is sometimes lower than existing baselines, computational complexity can be significantly lower than baselines depending on the selected query size.

**Questions:**

1. How was $Q$ (the number of latent queries) selected? Intuitively, 4-8 seems like a very small number given that each query can be interpreted as a sort of (dynamic) spatial filter, and that there are potentially many more brain sources that may be relevant to the downstream tasks. I understand that this is in fact an important tradeoff, as the larger $Q$ becomes, the higher the computational complexity becomes (if this gets on the same order as the channel dimension, the computational complexity benefits may actually disappear). Exploring, or at least discussing, the tradeoff between $Q$ vs.$E$ would be useful to understand this point.
2. The layouts seen during pretraining are in fact very consistent: only three montages were seen (20, 22, 29 channels). This, along with the lower performance on the SEED-V dataset with 62 channels, limits the impact of the study, as the appeal of such an approach is that very different montages can be handled in a unified manner. Exposing the model to more diverse channel configurations, e.g. by adding datasets with different montages to the pretraining set, or by randomly dropping out channels during training, could help generalization on new unseen montages. Also, testing on more varied electrode layouts would be important to support the conclusion (for instance, very sparse layouts for e.g. sleep data, and dense layouts from research-grade EEG systems).
3. If I understand correctly, the decoder of Section 3.2 requires knowing the number of channels of the input. Does this mean a separate decoder is required for each montage/dataset, or can these be reused between datasets? Of note, $E_{dec}$ is used to refer to both the decoder queries and the output of the cross-attention at line 163.

**Ethical Concerns:**

["NO or VERY MINOR ethics concerns only"]

**Final Justification:**

The submission presents a relevant EEG foundation model architecture with higher computational efficiency than existing models. The model is below the SOTA on three out of four tasks, but downstream performance remains generally competitive. As my concerns were appropriately addressed during the discussion period, notably on the importance of computational efficiency considerations, I raise my original score.

**Limitations:**

Yes.

**Quality:**

2

**Strengths And Weaknesses:**

Strengths

* Quality: The experimental setup, as in other EEG foundation model papers, relies on relevant downstream tasks to evaluate the usefulness of the representations learned during pretraining. This overall setup is sound, as is the comparison of theoretical computational complexity of the different existing models.
* Clarity: The paper is overall clearly written. The problem that is being solved is clearly exposed (varying electrode layouts and quadratic computational complexity), and the channel unification module is clearly explained. Most details necessary to reproduce the paper are described.
* Significance: The results are likely to have some impact on the community, as EEG foundation models are still being actively developed, and solutions to make them sensor layout-agnostic and reduce their computational complexity may facilitate scaling and usability in downstream scenarios.
* Originality: This work builds upon previous EEG foundation model work that are well cited and contextualized in the manuscript. However a relevant reference appears to be missing (see Weaknesses).

Weaknesses

* Quality:
    * The ablation studies of Section 4.4 (learned queries vs. fixed regions, query specialization loss) suggest that the change in performance is very small (3e-3 to 7e-3). These changes are on the same order as the reported standard deviation over a few seeds, and seem to actually yield improved performance on one of the metrics (TUAR AUC-PR). Drawing conclusions from these experiments is therefore difficult, and I would suggest increasing downstream task coverage before confirming these observations.
    * The layouts seen during pretraining are in fact very consistent, which limits the impact of the results (see Q2).
* Clarity: a few points remained unclear from the manuscript:
    * $A_{affinity}$ in the query specialization loss (line 172) is not defined.
    * It is not clear whether “multiclass classification” (line 189) refers to a multiclass classification problem, or actually to a “multilabel classification” problem, i.e. where each label is treated as a separate classification problem.
    * Preprocessing: it is not clear how the data can be converted to a bipolar montage (line 196).
    * How are the FLOPs of Figure 2 computed?
* Significance: Despite downstream performance being on a similar scale as existing EEG foundation models, the proposed approach underperforms similar-scale or smaller baseline models on three out of four downstream tasks. This limits the impact of the claims as it is not yet clear how important computational complexity constraints should be a limiting factor in EEG foundation model use cases (vs. performance).
* Originality:
    * The work of Saeed et al. (2020) is similar to the proposed approach, i.e. using learned query cross-attention to project input EEG channels to a fixed space (however without channel position information, and using a purely supervised learning task) and is tested on similar datasets. The differences with this work should be highlighted in the manuscript.
    * I would have expected BIOT to be included in the analysis of computational complexity, as it uses a linear attention mechanism, and still outperforms a larger LUNA model on at least one downstream task.

Saeed et al., 2020. Learning from Heterogeneous EEG Signals with Differentiable Channel Reordering, http://arxiv.org/abs/2010.13694.

---

> ### Author Rebuttal · Authors · 2025-07-30
>
> We thank the reviewer for their constructive and helpful feedback! You can find our question-wise responses below.
>
> ### Weakness-1: Conclusiveness of Ablation Studies
>
> We believe that these design choices have both quantitative and practical benefits:
>
> * Learned Queries vs. Fixed Regions: The primary advantage of using learned queries is not just the performance boost, but their practicality and flexibility. Our approach is fully data-driven and does not require any prior anatomical knowledge or mappings of electrodes to brain regions. This is a significant advantage over methods that rely on such priors, which can be overly simplistic, as brain regions are highly interconnected, or unavailable for non-standard montages.
> * Query Specialization Loss: The goal of this loss is to foster a more structured and interpretable latent space. By encouraging diversity, the loss pushes each query to learn a distinct spatial profile, effectively acting as an emergent, data-driven basis function for spatial unification. This specialization is visualized in Figure 4, where queries focus on different spatial regions.
>
> While we acknowledge the performance gains are on the same order as the standard deviation, it is important to note that the full model demonstrates a consistent positive trend. It achieves the highest AUROC on both TUAB and TUAR compared to its ablated counterparts. We believe that this consistent improvement in metrics, when combined with the significant practical benefits described above, provides a strong justification for our design choices.
>
> ***
>
> ### Weakness-2: Clarity of Technical Details
>
> 1.  **$A_{affinity}$ Definition**: We apologize for this omission. The term **$A_{affinity}$** refers to the **attention score matrix** from the Channel-Unification Module's cross-attention layer. This matrix has a shape of $(B \cdot S) \times Q \times C$ and represents the relationship between each of the $Q$ queries and $C$ input channels. We will add this precise definition in the revised manuscript.
>
> 2.  **Multiclass vs. Multilabel**:  This is indeed a "multiclass" problem. Each segment is assigned a single artifact label from a set of five distinct classes, similar to the EEGFormer setup. We will clarify this wording.
>
> 3.  **Bipolar Montage Conversion**: This is a standard EEG preprocessing technique where new virtual channels are created by taking the voltage difference between pairs of adjacent electrodes. We used the "double banana" method described in the [dataset description](https://isip.piconepress.com/publications/reports/2020/tuh_eeg/electrodes/electrodes_and_channels_v30.docx). We will clarify this.
>
> 4.  **FLOPs Calculation**: The FLOPs reported in Figure 2 were computed empirically using the `fvcore` library. We calculated the total GFLOPs for a single forward pass using both the encoder and decoder parts of these models.
>
> ***
>
> ### Weakness-3: Performance vs. Baselines and Significance
>
> We appreciate the reviewer's careful analysis of our results. While LUNA does not universally outperform all baselines on every metric, its primary contribution is providing a favorable accuracy-efficiency trade-off and a scalable architecture that is inherently topology-agnostic.
>
> We achieve state-of-the-art results on both the TUAR and TUSL benchmarks. On the tasks where performance is slightly lower (TUAB, SEED-V), LUNA offers substantial computational advantages. We want to highlight that models with less number of parameters might end up requiring more resources than LUNA, e.g., CBraMod. We believe computational efficiency to be an important aspect of EEG model development, as these models can potentially be deployed on edge devices for real-time monitoring.
>
> ***
>
> ### Weakness-4: Missing References and Baseline Comparisons
>
> We are very grateful to the reviewer for bringing the work of **Saeed et al. (2020)** to our attention and apologize for this oversight. It is indeed a relevant piece of work. We will add it to our Related Work section and highlight the key differences, such as LUNA being a self-supervised foundation model that focuses on efficiency.
>
> Regarding BIOT, this is an excellent suggestion. It can serve as an important benchmark for efficiency. We also added BIOT to our computational complexity analysis, and you can see a snippet of the resource usages below:
>
> #### Scaling with Patch Count
>
> | Model | Number of Patches | FLOPs (G) | Memory (MiB) |
> |:---|:---:|:---:|:---:|
> | **BIOT** | 2000 | 1143 | 3534 |
> | LUNA-Base | 2000 | 253 | 1835 |
> | LUNA-Large | 2000 | 1073 | 2969 |
> | LUNA-Huge | 2000 | 6570 | 5549 |
> | | | | |
> | **BIOT** | 3000 | 1714 | 5231 |
> | LUNA-Base | 3000 | 478 | 3873 |
> | LUNA-Large | 3000 | 1886 | 6041 |
> | LUNA-Huge | 3000 | 11035 | 9685 |
> | | | | |
> | **BIOT** | 4000 | 2286 | 6931 |
> | LUNA-Base | 4000 | 768 | 6678 |
> | LUNA-Large | 4000 | 2884 | 10265 |
> | LUNA-Huge | 4000 | 16286 | 15344 |
>
> #### Scaling with Channel Count
>
> | Model | Number of Channels | FLOPs (G) | Memory (MiB) |
> |:---|:---:|:---:|:---:|
> | **BIOT** | 6000 | 3117 | 9270 |
> | LUNA-Base | 6000 | 18 | 1571 |
> | LUNA-Large | 6000 | 43 | 2410 |
> | LUNA-Huge | 6000 | 112 | 4198 |
> | | | | |
> | **BIOT** | 7000 | 3637 | 10791 |
> | LUNA-Base | 7000 | 20 | 1826 |
> | LUNA-Large | 7000 | 49 | 2777 |
> | LUNA-Huge | 7000 | 122 | 4680 |
> | | | | |
> | **BIOT** | 8000 | 4156 | 12307 |
> | LUNA-Base | 8000 | 23 | 2069 |
> | LUNA-Large | 8000 | 55 | 3139 |
> | LUNA-Huge | 8000 | 133 | 5163 |
> | | | | |
>
> As can be seen from the table, BIOT is a more efficient model compared to LaBraM and CBraMod, but it still fails to scale better than LUNA when the number of patches/channels is increased. Moreover, BIOT uses fixed positional encodings for the time points and also for channels. Therefore, it is not topology-agnostic in the same way as LUNA is, and it is more limited.
> ***
>
> ### Question-1: Selection of Q
> The number of latent queries, `Q`, was determined empirically, guided by the goal of finding an optimal balance between model performance and computational efficiency. Each query can be interpreted as a learned, dynamic spatial filter, and the trade-off between the number of these filters (`Q`) and their representational capacity (the embedding dimension, `E`) is crucial.
>
> To provide a concrete answer and explore this trade-off, we conducted an ablation study on the LUNA-Base architecture. We kept the total latent feature dimension constant (`Q` × `E` = 256) while varying the two factors. The results are presented below.
> | Model Variant (`Q` x `E`) | #Queries(`Q`) | Embed Dim(`E`) | TUAB AUROC  | TUAB AUPRC  | TUAR AUROC  | TUAR AUPRC  | TUSL AUROC  | TUSL AUPRC  |
> |:-----------------------------:|:------------------:|:------------------:|:--------------------:|:--------------------:|:--------------------:|:--------------------:|:--------------------:|:--------------------:|
> | **LUNA-Base (4 x 64)**        | **4**              | **64**             | **0.887**      | **0.895**     | **0.902**      | 0.495    | **0.767**      | **0.301**    |
> | LUNA-Variant (2 x 128)        | 2                  | 128                | 0.885     | 0.890     | 0.885     | 0.501      | 0.759      | 0.288      |
> | LUNA-Variant (8 x 32)         | 8                  | 32                 | 0.884     | 0.892      | 0.899      | **0.505**     | 0.766      | 0.284      |
> | LUNA-Variant (16 x 16)        | 16                 | 16                 | 0.874      | 0.881      | 0.866      | 0.487      | 0.757      | 0.272      |
>
> ***
> The analysis reveals two key insights into the trade-off:
>
> 1.  A larger number of queries is not always better. The performance of the `16 x 16` variant is the lowest. This suggests that increasing `Q` at the expense of `E` is detrimental. While having more spatial filters is appealing, they are ineffective if they lack the individual embedding capacity (`E`) to learn rich representations. The queries might become too simplistic to capture meaningful patterns.
>
> 2.  A minimum number of diverse queries is necessary. The `2 x 128` variant, which uses only two highly expressive queries, performs slightly worse than the balanced `4 x 64` configuration on most tasks. This suggests that having too few queries can create an information bottleneck, even if each query is powerful. The model may struggle to simultaneously represent the diverse spatial patterns needed for complex downstream tasks.
>
> ### Question-2: Diversity of Pre-Training Montages
> This is a key limitation of our current study, which we acknowledge in the paper. The available large-scale public datasets are not topologically very diverse. This lack of diversity during pre-training is the likely reason for the performance gap on the unseen 62-channel SEED-V montage.
>
> We thank the reviewer for the excellent suggestions on how to mitigate this. Exposing the model to more diverse configurations during pre-training, like a random channel dropout strategy, is a promising direction for future work.
>
> ***
>
> ### Question-3: Decoder Design and Reusability
> 1. We initialize a learnable decoder query for each distinct electrode in the pre-training dataset. The reconstruction head used for pre-training retrieves the corresponding decoder queries from the learnable set for the input channels.
> 2. This reconstruction head is only used during pre-training to capture channel-specific features required to reconstruct each channel. If different datasets include shared electrodes, they can indeed be reused. For all downstream fine-tuning tasks, this head is discarded and replaced with a task-specific classification head. This classification head uses a single aggregation query and is completely independent of the original channel count, $C$. Therefore, the pre-trained LUNA encoder is fully reusable and montage-agnostic.
> 3. We apologize for the confusing reuse of the variable $E_{dec}$. We will rename a variable in the manuscript to clarify.

---

> > ### Comment · Reviewer_44Ri · 2025-08-05
> >
> > Thank you to the authors for the clarifications and additional analyses.
> >
> > 1. Conclusiveness of ablation studies
> >
> > I would argue that the variation in performance here is too small to be reliably described as a “performance boost”, unless demonstrated by statistical significance testing. Therefore, I would recommend to focus on the practical aspect of the design choices rather than the variation in performance, for instance at line 237: “[...] query diversity improves robustness, especially for complex artifacts”.
> >
> > 3. Performance vs. baselines
> >
> > I agree with the authors that computational efficiency is an important dimension to consider when designing EEG foundation models that can be deployed in real-time applications. What is unclear to me is whether existing SOTA models, e.g. CBraMod, are already too complex for these applications.
> >
> > To drive this point home, it would be interesting to give the reader a point of reference and provide available FLOPs/memory for a standard edge computing device that could reasonably be used with real-time EEG.

---

> > > ### Author Response · Authors · 2025-08-05
> > >
> > > We are very grateful to the reviewer for this insightful feedback. Your points will help us significantly improve the clarity and impact of our paper.
> > > ### 1. On the Conclusiveness of Ablation Studies
> > > We agree that our initial phrasing of a "performance boost" was too strong, given that the variation is within the margin of error. Your suggestion to focus on the practical benefits is an excellent recommendation, and we will revise the manuscript accordingly.
> > > Our revised argument in Section 4.4 will state that the primary value of the query specialization loss is not a statistically significant metric increase, but its role as a structural regularizer. We will focus on how the learned queries form a diverse and non-redundant basis set, demonstrated by the visualization of the queries.
> > >
> > > ### 2. On Grounding the Significance of Computational Efficiency
> > > We can analyze the feasibility of deploying a model like CBraMod versus LUNA on a realistic, power-efficient edge device by imaging a system-in-package architecture.
> > >
> > > While a simple microcontroller's on-chip memory is too small, the standard industrial solution for embedded AI is a System-in-Package (SiP). This design pairs a power-efficient core (like an ARM Cortex-M4 or the AI-specialized GAP9) with larger external RAM. Based on existing industrial products (e.g., from Octavo Systems or Microchip), a realistic system for this task would have an 8 MB to 32 MB RAM budget.
> > >
> > > Let's analyze the models against this budget:
> > > 1. SOTA Model: CBraMod (Varying number of parameters)
> > > Memory: The core issue with CBraMod's design is its O(C²) computational complexity in the spatial attention mechanism. For high-density EEG (e.g., 64+ channels), the memory required for intermediate activations and the number of operations (FLOPs) grow quadratically. This makes it extremely challenging to run in real-time on a power-constrained chip like a GAP9, which has a finite compute budget. Our analysis of the official CBraMod codebase also reveals a crucial limitation not mentioned in their paper: its decoder's architecture is hard-coded to a fixed input shape. It flattens the entire sequence (num_channels * num_patches * embed_size) before making a prediction. This means a model trained for 22 channels is architecturally incapable of processing a 62-channel input at deployment. Therefore, different-sized inputs will also require drastically different-sized models. The biggest CBraMod reported in the paper is for the TUAB dataset with 69M parameters, which requires ~138 MB (at 16-bit precision), larger than our entire RAM budget.
> > > 2. Our Model: LUNA-Base (7M parameters)
> > > Memory: LUNA-Base requires only ~14 MB for its weights. This fits comfortably within our 8-32 MB RAM budget, leaving space for the operating system and the dynamic activation maps needed during inference.
> > > Compute (FLOPs/OPS): The GAP9 chip can theoretically deliver up to 50 Giga-Operations per second (GOPS). LUNA's efficient architecture results in a low number of GFLOPs per inference (see Figure 2), making real-time processing plausible on this hardware.

---

> > > > ### Comment · Reviewer_44Ri · 2025-08-07
> > > >
> > > > Thank you for the analysis of deployment feasibility, which highlights the importance of the computational efficiency considerations discussed in the paper. My concerns have been appropriately addressed.

---

### Official Review · Reviewer_GV5R · 2025-07-02

**Clarity:** 3
**Significance:** 4
**Originality:** 3
**Rating:** 5
**Confidence:** 5

**Summary:**

This paper introduces a self-supervised foundation model designed to tackle the topological heterogeneity of EEG recordings (variations in channel configurations) during large-scale training. The method first segments each input sample into non-overlapping, single-channel patches and applies both temporal and frequency embeddings to these patches. A channel-unification module, built on a cross-attention mechanism, then compresses the variable channel sets into a fixed channel dimension. The unified representation is reshaped along the temporal axis and fed into a patch-wise temporal encoder. Self-supervised pre-training is performed through masked-patch reconstruction. The model is pre-trained on the TUEG and Siena Scalp EEG databases, and then fine-tuned on four downstream datasets, achieving state-of-the-art performance.

**Questions:**

See weakness.

**Ethical Concerns:**

["NO or VERY MINOR ethics concerns only"]

**Final Justification:**

I kept my score for 5.

**Limitations:**

See the conclusion part of the paper.

**Quality:**

4

**Strengths And Weaknesses:**

**Strengths**: 1) The manuscript is clearly written and easy to follow. The authors articulate a compelling motivation: topological heterogeneity across EEG datasets hampers the training of large-scale foundation models. 2) The proposed embedding module jointly encodes temporal, spatial, and spectral information—three pillars of effective EEG representation learning. 3) The channel-unification module built on cross-attention is intuitive and, in principle, broadly applicable to heterogeneous-channel scenarios. 4) The cross-attention mechanism is again utilized during reconstruction, and the downstream task is simplified and effective, thereby avoiding the need for different projection heads for various downstream datasets. 4) The paper presents a thoughtful analysis of complexity. By introducing a two-stage attention scheme—first unifying channels, then applying temporal self-attention—the computational cost is reduced from $S \times C$ to $S$, where $S$ is the number of patches per channel and $C$ the number of channels.

**Weaknesses**: 1) For a fair evaluation, the experiments on TUSL and TUAR should adopt subject-independent splits rather than random sample-level splits. 2) In Algorithm 1 (line 148), should the tensor $Q_{\text{learn}}$ have shape $\mathbb{R}^{B \times Q \times E}$ to remain symmetric with $E_{\text{dec}}$ on line 163 ($\mathbb{R}^{B \times C \times E}$)? Please clarify if my understanding is incorrect.

---

> ### Author Rebuttal · Authors · 2025-07-30
>
> We thank the reviewer for their constructive and helpful feedback! You can find our question-wise responses below.
> ### Weakness-1: Fairness of Evaluation Splits for TUAR/TUSL
>
> > For a fair evaluation, the experiments on TUSL and TUAR should adopt subject-independent splits rather than random sample-level splits.
>
> We agree that subject-independent evaluation is the gold standard for assessing clinical generalization, and we appreciate the feedback.
>
> Our decision to use a sample-level split for the TUAR and TUSL benchmarks was a deliberate one, to ensure a direct and fair comparison with the established previous literature. The evaluation protocols in recent leading works that we compare against, such as EEGFormer, have consistently used this random splitting mechanism. Adopting a different, subject-independent split would have made our results fundamentally incomparable to theirs, making it impossible to fairly assess the performance gains of our model relative to prior art. We would like to run all the baselines on a subject-independent split and hope to include this in future work.
>
> ### Weakness-2: Clarification on Tensor Shapes
>
> > In Algorithm 1 (line 148), should the tensor $Q_{learn}$ have shape $B \times Q \times E$ to remain symmetric with $E_{dec}$ on line 163 ($B \times C \times E$)?
>
> The query tensor $Q_{learn}$ is a set of $Q$ vectors of shape $E$, so the learnable parameter shape size is indeed $Q \times E$. Similarly, $E_{dec}$ is a collection of $C$ vectors of size $E$. These tensors are repeated in the batch dimension as needed. In the channel-unification module, the queries are repeated in the batch dimension to have the shape $(B.S) \times Q \times E$, to be able to apply cross-attention with the input features of size $(B·S) \times C \times E$. We will clarify this distinction of the initial learnable tensors and their reshaped version in the final manuscript better to avoid any confusions.

---

> > ### Comment · Reviewer_GV5R · 2025-08-05
> >
> > Hi, thanks for the rebuttal and clarification. It may be a good idea to include subject-independent results in the appendix for future researchers as a reference in your camera-ready version. Again, this paper is well-written and designed, and I will keep my score.

---

### Note · Authors · 2025-08-15

We sincerely thank the Area Chair and reviewers for their invaluable feedback. We are excited that our productive discussion resulted in three reviewers recommending acceptance (ratings: 5, 4, 4). We provided a detailed rebuttal to the initial concerns of the fourth reviewer (rating: 3), but were unable to engage further as they did not respond during the discussion period.

Our work positions LUNA as a direct architectural solution to a critical bottleneck in EEG analysis: the dual challenge of topological heterogeneity and computational complexity. The reviewers acknowledged the importance of this problem and the novelty of our channel-unification approach.

The discussion period was crucial for strengthening our claims. Reviewers raised several valid points, and in direct response, we:
1.  **Improved Efficiency Claims:** Following suggestions to include more baselines, we conducted new experiments comparing LUNA against BIOT. The results empirically prove LUNA’s superior scaling for both channel and patch count, a core advantage of our architecture.
2.  **Provided Empirical Justification for Key Hyperparameters:** In response to questions about our choice of `Q` (the number of queries), we ran a new ablation study on the `Q vs. E` trade-off. This provided clear, data-driven evidence that our chosen configuration offers the best balance between representational capacity and model complexity.
3.  **Clarified the Role of the `L_spec` Loss:** Through productive dialogue, we refined our claims about this component. We ran statistical t-tests, which confirmed it does not provide a significant performance boost. Instead, we provided new visualizations (as requested) that demonstrate its true value as a structural regularizer that enforces a diverse, interpretable latent space.
4.  **Committed to Correcting All Omissions:** We are very grateful to the reviewers for their attention. In the final version, we will add the missing reference to Saeed et al., correct all identified typos, clarify confusing notations, and add explicit definitions for technical details like bipolar montage conversion.

We are confident that the final manuscript, strengthened by the feedback and our new results, thoroughly addresses all raised issues. Thank you for your time and consideration.

---

### Decision · Program_Chairs · 2025-09-17

**Decision:**

Accept (poster)

**Comment:**

This paper introduces a self-supervised foundation model for EEG data. The proposed model provides a direct architectural solution to a critical bottleneck in EEG analysis: the dual challenge of topological heterogeneity and computational complexity.

The reviewers acknowledged several strengths, including clear writing, a well-motivated solution, and a novel, intuitive architectural design. During the rebuttal, the authors addressed most of the concerns raised and committed to revising the paper accordingly. Although Reviewer jKJ9 did not actively participate in the discussion and only responded after the rebuttal, I believe the issues they raised were satisfactorily addressed in the author response. The other three reviewers expressed positive inclinations toward acceptance.

I encourage the authors to incorporate the reviewers’ suggestions in the final version to further strengthen the work.

Recommendation: Accept as poster.